# A High-Dimensional Statistical Method for Optimizing Transfer Quantities in Multi-Source Transfer Learning

Qingyue Zhang[1]*, Haohao Fu[1]*, Guanbo Huang[1]*, Yaoyuan Liang[1], Chang Chu[1],
Tianren Peng[1], Yanru Wu[1], Qi Li[1], Yang Li[1]†, Shao-Lun Huang[1]†

[1]Tsinghua Shenzhen International Graduate School, Tsinghua University

## Abstract

Multi-source transfer learning provides an effective solution to data scarcity in real-world supervised learning scenarios by leveraging multiple source tasks. In this field, existing works typically use all available samples from sources in training, which constrains their training efficiency and may lead to suboptimal results. To address this, we propose a theoretical framework that answers the question: what is the optimal quantity of source samples needed from each source task to jointly train the target model? Specifically, we introduce a generalization error measure based on K-L divergence, and minimize it based on high-dimensional statistical analysis to determine the optimal transfer quantity for each source task. Additionally, we develop an architecture-agnostic and data-efficient algorithm OTQMS to implement our theoretical results for target model training in multi-source transfer learning. Experimental studies on diverse architectures and two real-world benchmark datasets show that our proposed algorithm significantly outperforms state-of-the-art approaches in both accuracy and data efficiency. The code is available at `https://github.com/zqy0126/OTQMS`.

## 1 Introduction

Nowadays, various machine learning algorithms have achieved remarkable success by leveraging large-scale labeled training data. However, in many practical scenarios, the limited availability of labeled data presents a significant challenge, where transfer learning emerges as an effective solution [33]. Transfer learning aims to leverage knowledge from tasks with abundant data or being well-trained, known as the source tasks, to improve the performance of a new learning task, known as the target task. Given its numerous applications, transfer learning has gained wide popularity and seen success in a variety of fields, such as computer vision [28], natural language processing [24], recommendation systems [7] and anomaly detection [26]. Traditionally, transfer learning has focused on the transfer between a single source task and a target task. However, there is a growing emphasis on multi-source transfer learning, which leverages multiple source tasks to enhance the training of the target task [23].

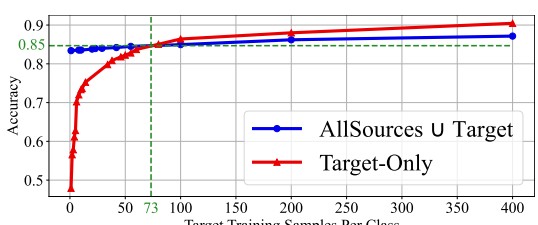

Figure 1: **More source samples does not always mean better performance.** Incorporating all source samples may bring negative impact, which is illustrated by the comparison of two strategies, using target task samples with all source samples (blue) and using target task samples only (red), evaluated on the equally divided 5-task CIFAR10 dataset. Theoretically, although incorporating more source samples reduces model variance by expanding the training data, the discrepancy between the source and target tasks introduces additional bias.

---

*Equal contribution.

†Corresponding authors: Yang Li (yangli@sz.tsinghua.edu.cn), Shao-Lun Huang(twn2gold@gmail.com)

39th Conference on Neural Information Processing Systems (NeurIPS 2025).

Table 1: **Comparison across matching-based transfer learning method,** based on whether they are tailored to multi-sources, have task generality, have shot generality, and require target labels. The '✓' represents obtaining the corresponding aspects, while '✗' the opposite. Task generality denotes the ability to handle various target task types, and shot generality denotes the ability to avoid negative transfer in different target sample quantity settings including few-shot and non-few-shot.

| Method | Multi-Source | Task Generality | Shot Generality | Target Label |
|---|---|---|---|---|
| MCW [12] | ✓ | ✗ | ✗ | Supervised |
| Leep [17] | ✗ | ✓ | ✓ | Supervised |
| Tong [25] | ✓ | ✗ | ✓ | Supervised |
| DATE [9] | ✓ | ✓ | ✓ | Unsupervised |
| H-ensemble [31] | ✓ | ✗ | ✗ | Supervised |
| OTQMS (Ours) | ✓ | ✓ | ✓ | Supervised |

In multi-source transfer learning, traditional methods usually jointly train the target model using all available samples from sources without selection [32, 22, 14]. This evidently poses a severe limitation to training efficiency, considering the vast number of available samples from various potential source tasks in real-world scenarios [19]. Moreover, directly assuming the use of all available samples seriously constrains their solution space, which possibly leads to suboptimal results, as illustrated in Figure 1. Therefore, it is critical to establish a theoretical framework to answer the question: *what is the optimal transfer quantity of samples from each source task needed in training the target task?*

In this work, we formulate a sample-based multi-source transfer learning problem as a parameter estimation problem. By employing high-dimensional statistical methods to analyze it, we establish a theoretical framework to determine the optimal transfer quantity for each source task. Specifically, we introduce the expectation of Kullback-Leibler (K-L) divergence between the true distribution of target task samples and the distribution learned from training samples as a measure of generalization error. This measure is then minimized in the asymptotic regime to derive the optimal transfer quantity of each source task. Building on this, we propose a practical algorithm, named **O**ptimal **T**ransfer **Q**uantities for **M**ulti-**S**ource Transfer learning (OTQMS), to implement our theoretical results for training multi-source transfer learning models. Notably, OTQMS is data-efficient and compatible with various model architectures, including Vision Transformer (ViT) and Low-Rank Adaptation (LoRA). It also demonstrates advantages in *task generality* and *shot generality* (as illustrated in Table 1), since we establish theoretical framework without restricting it to a specific target task type or limiting the range of target sample quantity. In summary, our main contributions are as follows:

a) We use high-dimensional statistics to analyze the parameter estimation problem formulated by the sample-based multi-source transfer learning problem. Based on this, a novel theoretical framework that optimizes transfer quantities is introduced.

b) Based on the framework, we propose OTQMS, an architecture-agnostic and data-efficient algorithm for target model training in multi-source transfer learning. In particular, we propose a dynamic strategy in OTQMS to alleviate the estimation error of transfer quantities caused by the scarcity of target task samples.

c) Experimental studies on few-shot multi-source transfer learning tasks, two real-world datasets and various model architectures demonstrate that OTQMS achieves significant improvements in both accuracy and data efficiency. In terms of accuracy, OTQMS outperforms state-of-the-art approaches by an average of $1.5\%$ on `DomainNet` and $1.0\%$ on `Office-Home`. In terms of data efficiency, OTQMS reduces the average training time by $35.19\%$ and the average sample usage by $47.85\%$ on `DomainNet`. Furthermore, extensive supplementary experiments demonstrate that OTQMS can facilitate both full model training and parameter-efficient training, and OTQMS is also applicable to multi-task learning tasks.

## 2 Related Work

This work is related to two main lines of research. The first is transfer learning theory, where we propose a measure based on K-L divergence as a novel measure of generalization error, different from

previously adopted measures [10, 2, 25, 1, 31]. The second is multi-source transfer learning, where most existing studies either utilize all samples from all source tasks or adopt task-level selection strategies [8, 22, 25, 31]. In contrast, our framework explicitly optimizes the transfer quantity from each individual source task. Other closely related work is discussed further in Appendix B.

## 3 Problem Formulation

Consider the transfer learning setting with one target task $\mathcal{T}$, and $K$ source tasks $\{\mathcal{S}_1, \ldots, \mathcal{S}_K\}$. The target task $\mathcal{T}$ is not restricted to a specific downstream task category. Generally, we formulate it as a parameter estimation problem under a distribution model $P_{X;\theta}$. For example, when $\mathcal{T}$ is a supervised classification task, $P_{X;\theta}$ corresponds to the joint distribution model of input features $Z$ and output labels $Y$, i.e., $X = (Z, Y)$. Our objective is to estimate the true value of $\underline{\theta}$, which corresponds to optimizing the neural network parameters for target task $\mathcal{T}$. Here, $\theta$ denotes 1-dimensional parameter, and $\underline{\theta}$ denotes high-dimensional parameter. Furthermore, we assume that the source tasks and the target task follow the same parametric model and share the same input space $\mathcal{X}$. Without loss of generality, we assume $\mathcal{X}$ to be discrete primarily for clarity of writing and to avoid redundancy, and our results can be readily extended to continuous spaces. The target task $\mathcal{T}$ has $N_0$ training samples $X^{N_0} = \{x_1, \ldots, x_{N_0}\}$ i.i.d. generated from some underlying joint distribution $P_{X;\underline{\theta}_0}$, where the parameter $\underline{\theta}_0 \in \mathbb{R}^d$. Similarly, the source task $\mathcal{S}_i$ has $N_i$ training samples $X^{N_i} = \{x_1^i, \ldots, x_{N_i}^i\}$ i.i.d. generated from some underlying joint distribution $P_{X;\underline{\theta}_i}$, where $i \in [1, K]$, and the parameter $\underline{\theta}_i \in \mathbb{R}^d$. In this work, we use the Maximum Likelihood Estimator (MLE) to estimate the true target task parameter $\theta_0$. Moreover, the following lemma characterizes the asymptotic behavior of MLE.

**Lemma 1.** *(Asymptotic Normality of the MLE) [29] When we use MLE only based on target task samples to estimate $\theta_0$, i.e.*

$$\hat{\underline{\theta}}_{MLE} = \arg\max_{\underline{\theta}} \frac{1}{N_0} \sum_{x \in X^{N_0}} \log P_{X;\underline{\theta}}(x), \tag{1}$$

*under appropriate regularity conditions, the following holds:*

$$\sqrt{N_0} \left( \hat{\underline{\theta}}_{MLE} - \underline{\theta}_0 \right) \xrightarrow{d} \mathcal{N} \left( 0, J(\underline{\theta}_0)^{-1} \right), \tag{2}$$

*where "$-1$" denotes the matrix inverse and the $J(\underline{\theta})$ is the Fisher information matrix defined as:*

$$J(\underline{\theta})^{d \times d} = \mathbb{E} \left[ \left( \frac{\partial}{\partial \underline{\theta}} \log P_{X;\underline{\theta}} \right) \left( \frac{\partial}{\partial \underline{\theta}} \log P_{X;\underline{\theta}} \right)^T \right]. \tag{3}$$

When we transfer $n_1, \ldots, n_K$ samples from the $\{\mathcal{S}_1, \ldots, \mathcal{S}_K\}$, where $n_i \in [0, N_i]$, we denote these training sample sequences as $X^{n_1}, X^{n_2}, \ldots, X^{n_K}$. Then, the MLE for multi-source transfer learning, is the parameter value that maximizes the empirical mean of the likelihood of samples from source tasks and target task, i.e.,

$$\hat{\underline{\theta}} = \arg\max_{\underline{\theta}} \frac{1}{N_0 + \sum_{i=1}^K n_i} \left( \sum_{x \in X^{N_0}} \log P_{X;\underline{\theta}}(x) + \sum_{i=1}^K \sum_{x \in X^{n_i}} \log P_{X;\underline{\theta}}(x) \right). \tag{4}$$

In this work, our goal is to derive the optimal transfer quantities $n_1^*, \ldots, n_K^*$ of source tasks $\mathcal{S}_1, \ldots, \mathcal{S}_K$ to minimize certain divergence measure $\mathcal{D}iv$ between the true distribution of target task $P_{X;\underline{\theta}_0}$ and the distribution $P_{X;\hat{\underline{\theta}}}$ learned from training samples, i.e.,

$$n_1^*, \ldots, n_K^* = \arg\min_{n_1, \ldots, n_K} \mathcal{D}iv(P_{X;\underline{\theta}_0}, P_{X;\hat{\underline{\theta}}}). \tag{5}$$

Besides, we provide the notations table in Appendix A.

## 4 Theoretical Analysis and Algorithm

In this section, we will first introduce a new K-L divergence based measure for the optimization problem in (5). Then, we will analyze it based on high-dimensional statistics to derive the optimal transfer quantities for both single-source and multi-source scenarios. Finally, we will develop a practical algorithm based on the theoretical framework.

**Definition 2.** *(The K-L divergence) [4] The K-L divergence $D\left(P\|Q\right)$ measures the difference between two probability distributions $P(X)$ and $Q(X)$ over the same probability space. It is defined as:*

$$D\left(P\|Q\right) = \sum_{x \in \mathcal{X}} p(x) \log \frac{p(x)}{q(x)}.$$

In this work, we apply the expectation of the K-L divergence between the true distribution model of the target task and the distribution model learned from training samples to measure the generalization error. Compared to other measures, the K-L divergence exhibits a closer correspondence with the generalization error as measured by the cross-entropy loss, .

Finally, our generalization error measure is defined as:

$$\mathcal{D}iv(P_{X;\theta_0}, P_{X;\hat{\theta}}) = \mathbb{E}\left[D\left(P_{X;\theta_0}\Big\|P_{X;\hat{\theta}}\right)\right]. \tag{6}$$

### 4.1 Single-Source Transfer Learning

The direct computation of the proposed K-L divergence based measure is challenging. Fortunately, we can show that the proposed measure is computable in the asymptotic regime using high-dimensional statistical analysis. To be specific, we can prove that the proposed measure directly depends on the Mean Squared Error (MSE) in the asymptotic regime, and the MSE can be calculated by an extension of Lemma 1. Therefore, we perform an asymptotic analysis of this measure.

In this section, we begin by presenting the theoretical results in Lemma 3 and Theorem 4 where the parameter is 1-dimensional, and subsequently extend them to the high-dimensional parameter setting in Proposition 5 and 6. To begin, we consider the setting with a target task $\mathcal{T}$ with $N_0$ samples generated from a model with 1-dimensional parameter.

**Lemma 3.** *(proved in Appendix C.1) Given a target task $\mathcal{T}$ with $N_0$ i.i.d. samples generated from a 1-dimensional underlying model $P_{X;\theta_0}$, where $\theta_0 \in \mathbb{R}$, and denoting $\hat{\theta}$ as the MLE (1) based on the $N_0$ samples, then the proposed measure (6) can be expressed as:*

$$\mathbb{E}\left[D\left(P_{X;\theta_0}\Big\|P_{X;\hat{\theta}}\right)\right] = \frac{1}{2N_0} + o\left(\frac{1}{N_0}\right). \tag{7}$$

The result of Lemma 3 demonstrates that when there is only one target task without any source task, the proposed measure is inversely proportional to the number of training samples. Next, we consider the transfer learning scenario where we have one target task $\mathcal{T}$ with $N_0$ training samples and one source task $\mathcal{S}_1$ with $N_1$ training samples. In this case, we aim to determine the optimal transfer quantity $n_1^* \in [1, N_1]$. To facilitate our mathematical derivations, we assume $N_0$ and $N_1$ are asymptotically comparable, and the distance between the parameters of the target task and source task is sufficiently small (*i.e.*, $|\theta_0 - \theta_1| = O(\frac{1}{\sqrt{N_0}})$). Considering the similarity of low-level features among tasks of the same type, this assumption is made without loss of generality and supported by related studies [20]. Furthermore, as demonstrated in subsequent analysis, our conclusions remain valid even in extreme cases where the distance between parameters is large.

**Theorem 4.** *(proved in Appendix C.2) Given a target task $\mathcal{T}$ with $N_0$ i.i.d. samples generated from a 1-dimensional underlying model $P_{X;\theta_0}$, and a source task $\mathcal{S}_1$ with $N_1$ i.i.d. samples generated from a 1-dimensional underlying model $P_{X;\theta_1}$, where $\theta_0, \theta_1 \in \mathbb{R}$ and $|\theta_0 - \theta_1| = O(\frac{1}{\sqrt{N_0}})$, $\hat{\theta}$ is denoted as the MLE (4) based on the $N_0$ samples from $\mathcal{T}$ and $n_1$ samples from $\mathcal{S}_1$, where $n_1 \in [0, N_1]$, then the proposed measure (6) can be expressed as:*

$$\frac{1}{2}\left(\underbrace{\frac{1}{N_0 + n_1}}_{\text{variance term}} + \underbrace{\frac{n_1^2}{(N_0 + n_1)^2}t}_{\text{bias term}}\right) + o\left(\frac{1}{N_0 + n_1}\right), \tag{8}$$

*where*

$$t \triangleq J(\theta_0)(\theta_1 - \theta_0)^2. \tag{9}$$

*Moreover, the optimal transfer quantity $n_1^*$ is*

$$n_1^* = \begin{cases} N_1, & \text{if } N_0 \cdot t \leq 0.5 \\ \min\left(N_1, \frac{N_0}{2N_0 t - 1}\right), & \text{if } N_0 \cdot t > 0.5 \end{cases}. \tag{10}$$

From (8), we observe that the proposed measure decreases as $N_0$ increases, which aligns with our intuition that utilizing all available target samples is beneficial. In addition, the trend of (8) with respect to $n_1$ is more complex. We plot (8) as a function of $n_1$ under two different regimes, determined by the value of $N_0 \cdot t$, as shown in Figure 2. Our goal is to explore the optimal value of $n_1^*$ to minimize (8).

- **Case 1** ($N_0 \cdot t \leq 0.5$)**:** The proposed measure monotonically decreases as $n_1$ increases. Obviously, the optimal point is $n_1^* = N_1$. This indicates that when the source task and the target task are highly similar, *i.e.*, when $t$ is small, an increase in the transfer quantity will positively impact the results.

- **Case 2** ($N_0 \cdot t > 0.5$)**:** The proposed measure first decreases and then increases as $n_1$ increases. It attaining its minimum at $n_1' = \frac{N_0}{2N_0 t - 1}$. It should be noted that when $t$ is large enough, the point $n_1'$ approaches $0$. This aligns with the intuition that when the discrepancy between the source and target tasks is substantial, avoiding transfer yields better results. Furthermore, if $n_1'$ exceeds $N_1$, we should utilize all $N_1$ samples, so $n_1^* = \min\left(N_1, \frac{N_0}{2N_0 t - 1}\right)$.

As the dimensionality of the model parameter $\theta$ increases to **higher dimensions**, we derive the following propositions.

**Proposition 5.** *(proved in Appendix C.3) In the case where the parameter dimension is d, i.e., $\underline{\theta}_0 \in \mathbb{R}^d$, with all other settings remaining the same as the Lemma 3, the proposed measure* (6) *can be expressed as*

$$\frac{d}{2N_0} + o\left(\frac{1}{N_0}\right). \tag{11}$$

**Proposition 6.** *(proved in Appendix C.4) In the case where the parameter dimension is d, i.e., $\underline{\theta}_0, \underline{\theta}_1 \in \mathbb{R}^d$, with all other settings remaining the same as the Theorem 4, the proposed measure* (6) *can be expressed as*

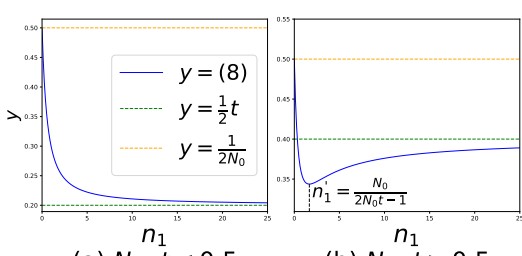

(a) $N_0 \cdot t \leq 0.5$    (b) $N_0 \cdot t > 0.5$

Figure 2: The function curve figures of (8) under different regimes determined by the value of $N_0 \cdot t$ (blue). The vertical axis denotes the value of proposed measure (8), while the horizontal axis denotes the variable $n_1$.

$$\frac{d}{2}\left(\frac{1}{N_0 + n_1} + \frac{n_1^2}{(N_0 + n_1)^2}t\right) + o\left(\frac{1}{N_0 + n_1}\right), \tag{12}$$

*where we denote*

$$t \triangleq \frac{(\underline{\theta}_1 - \underline{\theta}_0)^T J(\underline{\theta}_0)(\underline{\theta}_1 - \underline{\theta}_0)}{d}. \tag{13}$$

*In addition, t is a scalar, $J(\underline{\theta}_0)$ is $d \times d$ matrix, and $(\underline{\theta}_1 - \underline{\theta}_0)$ is a $d - dimensional$ vector, which is the element-wise subtraction of two d-dimensional vectors $\underline{\theta}_1$ and $\underline{\theta}_0$.*

Compared to Theorem 4, Proposition 6 exhibits a similar mathematical form, allowing us to derive the optimal transfer quantity through a similar approach. Furthermore, we observe that as the parameter dimension $d$ increases, the K-L error measure (12) increases. This suggests that for a complex model, knowledge transfer across tasks becomes more challenging, which is consistent with the findings in [25].

## 4.2   Multi-Source Transfer Learning

Consider the multi-source transfer learning scenario with $K$ source task $\{\mathcal{S}_1, \ldots, \mathcal{S}_K\}$ and one target task $\mathcal{T}$. We aim to derive the optimal transfer quantity $n_i^*$ of each source.

**Theorem 7.** *(proved in Appendix C.5) Given a target task $\mathcal{T}$ with $N_0$ i.i.d. samples generated from the underlying model $P_{X;\underline{\theta}_0}$, and K source tasks $\mathcal{S}_1, \ldots, \mathcal{S}_K$ with $N_1, \ldots, N_K$ i.i.d. samples generated from the underlying model $P_{X;\underline{\theta}_1}, \ldots, P_{X;\underline{\theta}_K}$, where $\underline{\theta}_0, \underline{\theta}_1, \ldots, \underline{\theta}_K \in \mathbb{R}^d$. $\hat{\underline{\theta}}$ is denoted as the MLE* (4) *based on the $N_0$ samples from $\mathcal{T}$ and $n_1, \ldots, n_K$ samples from $\mathcal{S}_1, \ldots, \mathcal{S}_K$, where*

$n_i \in [0, N_i]$. *Denoting* $s = \sum_{i=1}^{K} n_i$ *as the total transfer quantity, and* $\alpha_i = \frac{n_i}{s}$ *as the proportion of different source tasks, then the proposed measure* (6) *can be expressed as:*

$$\frac{d}{2}\left(\frac{1}{N_0 + s} + \frac{s^2}{(N_0 + s)^2}t\right) + o\left(\frac{1}{N_0 + s}\right). \tag{14}$$

*In* (14), $t$ *is a scalar denoted as*

$$t = \frac{\underline{\alpha}^T \Theta^T J(\underline{\theta}_0)\Theta\underline{\alpha}}{d}, \tag{15}$$

*where* $\underline{\alpha} = [\alpha_1, \ldots, \alpha_K]^T$ *is a K-dimensional vector, and* $\Theta^{d \times K} = [\underline{\theta}_1 - \underline{\theta}_0, \ldots, \underline{\theta}_K - \underline{\theta}_0]$.

According to Theorem 7, we can derive the optimal transfer quantities $n_1^*, \ldots, n_K^*$ by minimizing (14). Equivalently, we need to find the optimal total transfer quantity $s^*$ and the optimal proportion vector $\underline{\alpha}^*$ which minimize (14). The analytical solutions of $s^*$ and $\underline{\alpha}^*$ are difficult to acquire, and we provide a method to get their numerical solutions in Appendix E. Eventually, we can get the optimal transfer quantity of each source through $n_i^* = s^* \cdot \alpha_i^*$.

### 4.3 Practical Algorithm

Along with our theoretical framework, we propose a practical algorithm, OTQMS, which is applicable to sample-based multi-source transfer learning tasks, as presented in Algorithm 1. In OTQMS, when computing the optimal transfer quantities based on Theorem 7, we use the parameters of pretrained source models to replace $\underline{\theta}_1, \cdots, \underline{\theta}_K$. Considering that the source model can be trained using sufficient labeled data, it is reasonable to use the learned parameters as a good approximation of the true underlying parameters. In contrast, the number of target data in transfer learning is often insufficient, so it is difficult to accurately estimate the true parameter $\underline{\theta}_0$ using only the target data. Therefore, as shown in lines 5-14 of Algorithm 1, we adopt a **dynamic strategy**. Specifically, in the first epoch, we train a $\underline{\theta}_0$ using only the target data. This $\underline{\theta}_0$ is then used, along with Theorem 7, to determine the optimal transfer quantity from each source task, and we use **random sampling** to form a new resampled training dataset. Finally, we continue training $\underline{\theta}_0$ on this new dataset, and this procedure is repeated in each subsequent epoch to iteratively update the training dataset. This mechanism helps alleviate the inaccuracy of $\underline{\theta}_0$. In particular, we compute the matrix $J$ using the empirical mean of gradient outer products in line 10, with the loss function $\ell$ restricted to the negative log-likelihood, following a widely adopted approach known as the **empirical Fisher** [16, 18].

---

**Algorithm 1** OTQMS: Training

---

1: **Input:** Target data $D_{\mathcal{T}} = \{(z_{\mathcal{T}}^j, y_{\mathcal{T}}^j)\}_{j=1}^{N_0}$, source data $\{D_{S_k} = \{(z_{S_k}^j, y_{S_k}^j)\}_{j=1}^{N_k}\}_{k=1}^{K}$, model type $f_{\underline{\theta}}$ and its parameters $\underline{\theta}_0$ for target task and $\{\underline{\theta}_k\}_{k=1}^{K}$ for source tasks, parameter dimension $d$.      // $z$ represents the feature and $y$ represents the label

2: **Parameter:** Learning rate $\eta$.

3: **Initialize:** randomly initialize $\underline{\theta}_0$, initialize $\{\underline{\theta}_k\}_{k=1}^{K}$ with parameters of pretrained source models.

4: **Output:** a well-trained $\underline{\theta}_0$ for target task model $f_{\underline{\theta}_0}$.

5:    $D_{train} \leftarrow D_{\mathcal{T}}$        // Initialize the trainning dataset by target task samples

6: **repeat**        // Use dynamic strategy to train the target task

7:      $\mathcal{L}_{train} \leftarrow \frac{1}{|D_{train}|} \sum_{(y^j, z^j) \in D_{train}} \ell\left(y^j, f_{\underline{\theta}_0}(z^j)\right)$

8:      $\underline{\theta}_0 \leftarrow \underline{\theta}_0 - \eta \nabla_{\underline{\theta}_0} \mathcal{L}_{train}$

9:      $\Theta \leftarrow [\underline{\theta}_1 - \underline{\theta}_0, \ldots, \underline{\theta}_K - \underline{\theta}_0]^T$

10:     $J(\underline{\theta}_0) \leftarrow \frac{1}{|D_{\mathcal{T}}|} \sum_{(z^j, y^j) \in D_{\mathcal{T}}} \nabla_{\underline{\theta}_0} \ell\left(y^j, f_{\underline{\theta}_0}(z^j)\right) \left(\nabla_{\underline{\theta}_0} \ell\left(y^j, f_{\underline{\theta}_0}(z^j)\right)\right)^\top$

11:     $(s^*, \underline{\alpha}^*) \leftarrow \underset{(s, \underline{\alpha})}{\arg\min} \frac{d}{2}\left(\frac{1}{N_0 + s} + \frac{s^2}{(N_0 + s)^2} \frac{\underline{\alpha}^T \Theta^T J(\underline{\theta}_0)\Theta\underline{\alpha}}{d}\right)$

12:     $D_{source} \leftarrow \bigcup_{k=1}^{K} \left\{D_{S_k}^* \Big| D_{S_k}^* \overset{\text{rand}}{\subseteq} D_{S_k}, |D_{S_k}^*| = s^* \alpha_k^*\right\}$

13:     $D_{train} \leftarrow D_{source} \bigcup D_{\mathcal{T}}$        // Update the trainning dataset

14: **until** $\underline{\theta}_0$ converges;

---

# 5 Experiments

## 5.1 Experiments Settings

**Benchmark Datasets.** `DomainNet` contains 586,575 samples of 345 classes from 6 domains (*i.e.*, **C**: Clipart, **I**: Infograph, **P**: Painting, **Q**: Quickdraw, **R**: Real and **S**: Sketch). `Office-Home` benchmark contains 15588 samples of 65 classes, with 12 adaptation scenarios constructed from 4 domains: Art, Clipart, Product and Real World (abbr. **Ar**, **Cl**, **Pr** and **Rw**). `Digits` contains four-digit sub-datasets: MNIST(mt), Synthetic(sy), SVHN(sv) and USPS(up), with each sub-dataset containing samples of numbers ranging from 0 to 9.

**Implementation Details.** We employ the ViT-Small model [30], pre-trained on ImageNet-21k [5], as the backbone for all datasets. The Adam optimizer is employed with a learning rate of $1e^{-5}$. We allocate 20% of the dataset as the test set, and report the highest accuracies within 5 epoch early stops in all experiments. Following the standard few-shot learning protocol, the training data for k-shot consists of k randomly selected samples per class from the target task. All experiments are conducted on Nvidia A800 GPUs.

**Baselines.** For a general performance evaluation, we take SOTA works under similar settings as baselines. The scope of compared methods includes: 1) Unsupervised Methods: MSFDA [21], DATE [9], M3SDA [19]. 2) Few-Shot Methods Based on Model(Parameter)-Weighting: H-ensemble [31], MCW [12]. 3) Few-Shot Methods Based on Sample: MADA [32], WADN [22] 4) Source Ablating Methods: Target-Only, Single-Source-Avg/Single-Source-Best (average/best performance of single-source transfer), AllSources ∪ Target (all source & target data in multi-source transfer).

Note that MADA [32] leverages all unlabeled target data in conjunction with a limited amount of labeled target data, which is a hybrid approach combining unsupervised and supervised learning. Due to the page limit, we provide detailed information on the experimental design and the results of an experiment adapted to the WADN settings on the `Digits` dataset in Appendix D.2.

Table 2: **Multi-Source Transfer Performance on DomainNet and Office-Home.** The arrows indicate transfering from the rest tasks. The highest/second-highest accuracy is marked in Bold/Underscore form respectively.

| Method | Backbone | DomainNet | | | | | | | Office-Home | | | | |
|---|---|---|---|---|---|---|---|---|---|---|---|---|---|
| | | →C | →I | →P | →Q | →R | →S | Avg | →Ar | →Cl | →Pr | →Rw | Avg |
| **Unsupervised-all-shots** | | | | | | | | | | | | | |
| MSFDA[21] | ResNet50 | 66.5 | 21.6 | 56.7 | 20.4 | 70.5 | 54.4 | 48.4 | 75.6 | 62.8 | 84.8 | 85.3 | 77.1 |
| DATE[9] | ResNet50 | - | - | - | - | - | - | - | 75.2 | 60.9 | **85.2** | 84.0 | 76.3 |
| M3SDA[19] | ResNet101 | 57.2 | 24.2 | 51.6 | 5.2 | 61.6 | 49.6 | 41.5 | - | - | - | - | - |
| **Supervised-10-shots** | | | | | | | | | | | | | |
| *Few-Shot Methods:* | | | | | | | | | | | | | |
| H-ensemble[31] | ViT-S | 53.4 | 21.3 | 54.4 | 19.0 | 70.4 | 44.0 | 43.8 | 71.8 | 47.5 | 77.6 | 79.1 | 69.0 |
| MADA[32] | ViT-S | 51.0 | 12.8 | 60.3 | 15.0 | **81.4** | 22.7 | 40.5 | **78.4** | 58.3 | 82.3 | 85.2 | 76.1 |
| MADA[32] | ResNet50 | 66.1 | 23.9 | 60.4 | 31.9 | 75.4 | 52.5 | 51.7 | 72.2 | 64.4 | 82.9 | 81.9 | 75.4 |
| MCW[12] | ViT-S | 54.9 | 21.0 | 53.6 | 20.4 | 70.8 | 42.4 | 43.9 | 68.9 | 48.0 | 77.4 | **86.0** | 70.1 |
| WADN[22] | ViT-S | 68.0 | 29.7 | 59.1 | 16.8 | 74.2 | 55.1 | 50.5 | 60.3 | 39.7 | 66.2 | 68.7 | 58.7 |
| *Source-Ablation Methods:* | | | | | | | | | | | | | |
| Target-Only | ViT-S | 14.2 | 3.3 | 23.2 | 7.2 | 41.4 | 10.6 | 16.7 | 40.0 | 33.3 | 54.9 | 52.6 | 45.2 |
| Single-Source-Avg | ViT-S | 50.4 | 22.1 | 44.9 | 24.7 | 58.8 | 42.5 | 40.6 | 65.2 | 53.3 | 74.4 | 72.7 | 66.4 |
| Single-Source-Best | ViT-S | 60.2 | 28.0 | 55.4 | 28.4 | 66.0 | 49.7 | 48.0 | 72.9 | 60.9 | 80.7 | 74.8 | 72.3 |
| AllSources ∪ Target | ViT-S | 71.7 | 32.4 | 60.0 | 31.4 | 71.7 | 58.5 | 54.3 | 77.0 | 62.3 | 84.9 | 84.5 | 77.2 |
| OTQMS (Ours) | ViT-S | **72.8** | **33.8** | **61.2** | **33.8** | 73.2 | **59.8** | **55.8** | 78.1 | **64.5** | 85.2 | 84.9 | **78.2** |

## 5.2 Main Result

We evaluated our algorithm, OTQMS, alongside baseline methods on the few-shot multi-source transfer learning tasks using the `DomainNet` and `Office-Home` datasets. The quantitative results are summarized in Table 2. Since the unsupervised baselines are not designed for the supervised few-shot setting, we report their original results from the respective papers for reference. We make the following observations:

**Overall Performance.** In general, compared to baseline methods, OTQMS achieves the best performance on almost all the transfer scenarios on the two datasets. Specifically, OTQMS outper-

forms the state-of-the-art (AllSources $\cup$ Target) by an average of $1.5\%$ on `DomainNet` and $1.0\%$ on `Office-Home`.

**Data Speaks Louder Than Model Weighting.** It is worth noting that on both datasets, sample-based methods utilizing both target and source samples to jointly train the model, such as WADN, MADA and OTQMS, generally outperform model(parameter)-weighting approaches which construct the target model by weighting source models, such as H-ensemble and MCW. This observation suggests that sample-based approaches offer greater advantages over model-based methods, because they can fully leverage the relevant information from the source data for the target task.

**Take, But Only as Much as You Need.** Comparing results in Table 2 among Target-only, AllSources $\cup$ Target, and OTQMS, we observe that OTQMS achieves the best performance in both datasets by leveraging only a subset of data selected from all available sources based on model preference. This result validates our theory. By choosing the right quantities of samples from the source tasks, we could train the target model more accurately, and we give an analysis on the domain preference of transfer quantity in Appendix F. Furthermore, Figure 4 shows that OTQMS also significantly reduces the training time and sample usage, which validates its superiority in terms of data efficiency.

**Few-Shot Labels, Big Gains.** We make a comparison of the results of unsupervised and supervised methods. While other conditions remain the same, Table 2 demonstrates that even if unsupervised methods like MSFDA and M3SDA take all the target data into account (up to $1.3 \times 10^5$ samples on Real domain of `DomainNet`), their performance still falls short compared to the supervised methods, which rely on only a limited number of samples (3450 samples). This illustrates the importance of having supervised information in multi-source transfer learning.

## 5.3 Static vs. Dynamic Transfer Quantity

In our proposed Algorithm 1, we employ a "Dynamic" strategy that dynamically determines the optimal transfer quantities and updates the resampled dataset during the joint training of target task. To validate the effectiveness of this strategy, we conducted comparative experiments using the "Static-∗" methods. "Static-∗" methods first simulate the distribution of target on target dataset only, and different types of Static such as "Under, Exact and Over" stands for different fitting levels. In "Static-∗" methods, we

Table 3: Static vs. Dynamic Transfer Quantity in OTQMS on Office-Home.

| Method | Backbone | Office-Home | | | | |
| --- | --- | --- | --- | --- | --- | --- |
| | | $\rightarrow$Ar | $\rightarrow$Cl | $\rightarrow$Pr | $\rightarrow$Rw | Avg |
| *Supervised-10-shots:* | | | | | | |
| Static-Under | ViT-S | 77.0 | 62.3 | 84.9 | 84.5 | 77.2 |
| Static-Exact | ViT-S | 46.0 | 59.8 | 85.1 | 83.7 | 68.7 |
| Static-Over | ViT-S | 76.8 | 61.9 | 78.6 | 68.6 | 71.5 |
| Dynamic | ViT-S | 78.1 | 64.5 | 85.2 | 84.9 | 78.2 |

only compute the optimal transfer quantity once to make the resampled dataset, and evaluated on it until target model converges. The results on Table 3 demonstrate OTQMS using dynamic transfer quantity achieved the best performance.

## 5.4 Generality across Different Shot Settings

As discussed in the theoretical analysis of Theorem 4, our theoretical framework is applicable to any quantity of target samples. Therefore, OTQMS exhibits shot generality, enabling it to avoid negative transfer across different shot settings. To validate this, we increase the number of shots from 5 to 100 across methods including AllSources $\cup$ Target, Target-Only, and OTQMS. As shown in Figure 3, experimental results demonstrate that OTQMS consistently outperforms other approaches across all shot settings. This highlights the generality and scalability of OTQMS in terms of data utilization.

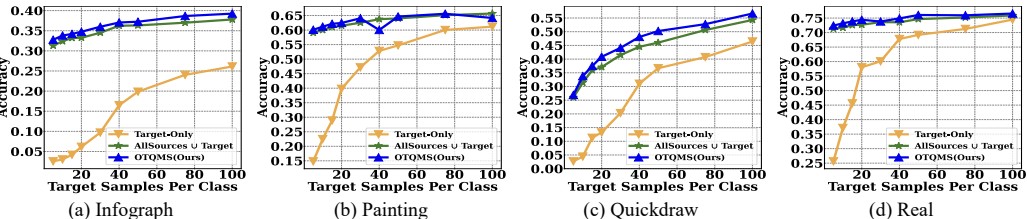

(a) Infograph     (b) Painting     (c) Quickdraw     (d) Real

Figure 3: Performance comparison with increasing target shots up to 100 per class on DomainNet dataset (I, P, Q and R domains). OTQMS (blue) outperforms other methods.

## 5.5 Data Efficiency Test

In this section, we demonstrate the advantage of OTQMS in terms of data efficiency. Specifically, we conduct experiments with MADA, AllSources ∪ Target, and OTQMS across different shot settings, and for each shot setting, we accumulate the total sample used and time consumed until the highest accuracy is reached. To better visualize the results, we present the average sample usage and training time across all shot settings in Figure 4. To be specific, OTQMS reduces the average training time by $35.19\%$ and the average sample usage by $47.85\%$ on `DomainNet`, compared to the AllSources ∪ Target method.

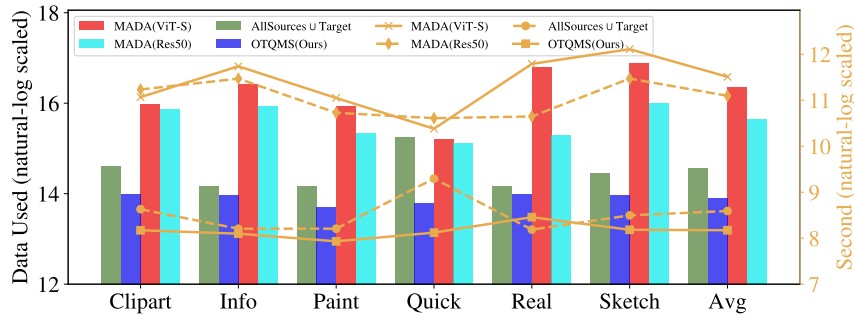

Figure 4: Data efficiency comparison of average sample usage and training time on `DomainNet` dataset, the left vertical axis represents the amount of sample usage, with green bars indicating AllSources ∪ Target data counts, blue bars about OTQMS, red bars about MADA(ViT-S) and azury bars about MADA(Res50), while the right orange vertical axis and lines represent training time.

## 5.6 Compatibility to Parameter-Efficient Training

To evaluate the applicability with parameter-efficient training, we used a ViT-Base model [6] integrated with the LoRA framework [11] as the backbone. This approach significantly reduces the number of trainable parameters for downstream tasks while ensuring model quality and improving training efficiency. In the experiments, we consider the trainable low-rank matrices and the classification head as the parametric model in our theoretical framework, while treating the remaining parameters as constants. Other experimental settings are the same as default. Experiments were conducted on the `Office-Home` dataset. The results of Table 6 in Appendix demonstrate that our method remains effective.

## 5.7 Compatibility to Multi-Task Learning

To demonstrate the applicability of our method to multi-task learning scenarios, we conduct experiments on the `Office-Home` dataset using the ViT-Small model. In multi-task learning, each task simultaneously serves as both a source and a target task. In the experiments, each task is treated as the target task in OTQMS in turn during training, while the transfer

Table 4: Multi-task performance on four tasks of Office-Home.

| Method | Backbone | Office-Home | | | | |
|---|---|---|---|---|---|---|
| | | Ar | Cl | Pr | Rw | Avg |
| Single-task | ViT-S | 66.7 | 62.3 | 87.8 | 68.6 | 71.4 |
| OTQMS | ViT-S | **81.7** | **76.0** | **88.6** | **87.5** | **83.5** |

quantities from all source tasks are computed in turn. This setup enables us to evaluate how effectively OTQMS leverages information across tasks. We compare the performance of our method against single-task training to evaluate its effectiveness in Table 4.

## 6 Conclusion

In this work, we propose a theoretical framework to determine the optimal transfer quantities in multi-source transfer learning. Our framework reveals that by optimizing the transfer quantity of each source task, we can improve target task training while reducing the total transfer quantity. Based on this theoretical framework, we develop an architecture-agnostic and data-efficient practical algorithm OTQMS for jointly training the target model. We evaluated the proposed algorithm through extensive experiments and demonstrated its superior accuracy and enhanced data efficiency.

## Acknowledge

The research is supported in part by National Key R&D Program of China under Grant 2021YFA0715202, the National Natural Science Foundation of China under Grants 62571296, the Shenzhen Science and Technology Program under KJZD20240903102700001, and the Natural Science Foundation of China (Grant 62371270).

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

# A Notations

| Symbol | Description |
|---|---|
| $\mathcal{T}$ | target task |
| $\{\mathcal{S}_1, \ldots, \mathcal{S}_K\}$ | source tasks |
| $N_0$ | quantity of target samples |
| $N_1 \cdots N_K$ | maximum sample quantity of each source |
| $n_1 \cdots n_K$ | transfer quantity of each source |
| $\theta$ | model parameter |
| $\underline{\theta}$ | vectorized model parameter |
| $\underline{\theta}_0$ | vectorized model parameter of target task |
| $\underline{\theta}_i, i \in [1, K]$ | vectorized model parameter of i-th source task, |
| $J(\theta)$ | Fisher information (scalar) of $\theta$ |
| $J(\underline{\theta})^{d \times d}$ | Fisher information (matrix) of d-dimensional $\underline{\theta}$ |
| $P_{X;\theta}$ | distribution of X parameterized by $\theta$ |
| $||\underline{x}||^2$ | l-2 norm of vector x |
| $\alpha_i$ | transfer proportion in multi-source case from i-th source task |
| $\underline{\alpha}$ | transfer proportion vector in multi-source case whose i-th entry is $\alpha_i$ |
| $s$ | total transfer quantity in multi-source case |
| $\hat{\theta}$ | estimator of $\theta$ |
| $E_{\hat{\theta}}$ | expectation of $\hat{\theta}$ |

Table 5: Notations

# B  Extended Related Work

## B.1  Transfer Learning Theory

Existing theoretical works can be categorized into two groups. The first group focuses on proposing measures to quantify the similarity between the target and source tasks. Within this group, some measures have been introduced, including $l_2$-distance [15], optimal transport cost [3], LEEP [17], Wasserstein distance [22], and maximal correlations [12]. This work belongs to the second group focusing on developing new generalization error measures. Within this group, the measures having been introduced include $f$-divergence [10], mutual information [2], $\mathcal{X}^2$-divergence [25], $\mathcal{H}$-score [1, 31]. However, the potential of K-L divergence as a generalization error measure has not been sufficiently explored.

## B.2  Multi-source Transfer Learning

Classified by the object of transfer, existing multi-source transfer learning methods mainly focus on two types: model transfer vs sample transfer [36]. Model transfer assumes there is one or more pre-trained models on the source tasks and transfers their parameters to the target task via fine-tuning [27]. This work focuses on the latter, which is based on joint training of the source task samples with those of the target task [32, 22, 14]. Classified by the strategy of transfer, existing methods mainly focus on two types: alignment strategy and matching-based strategy [35]. Alignment strategy aims to reduce the domain shift among source and target domains [13, 34, 14]. This work is more similar to the latter, focusing on determining which source domains or samples should be selected or assigned higher weights for transfer [8, 22, 25, 31]. However, most existing works either utilize all samples from all sources or perform task-level selection, whereas this work explores a framework that optimizes the transfer quantity of each source task. Moreover, many works are restricted to specific target tasks, such as classification, which limits their *task generality*. In addition, many studies are mainly applicable to few-shot scenarios, and may suffer from negative transfer in non-few-shot settings, which limits their *shot generality*, as illustrated in Table 1.

## C  Proofs

### C.1  Proof of Lemma 3

**Lemma 8.** *In the asymptotic case, the proposed measure* (6) *and the mean squared error have the relation as follows.*

$$\mathbb{E}\left[D\left(P_{X;\theta_0}\middle\|P_{X;\hat{\theta}}\right)\right]$$
$$= \frac{1}{2}J(\theta_0)MSE(\hat{\theta}) + o(\frac{1}{N_0}), \tag{16}$$

*Proof.* In this section, for the sake of clarity, we will write $\hat{\theta}$ in its parameterized form $\hat{\theta}(X^{N_0})$ when necessary, and these two forms are mathematically equivalent. By taking Taylor expansion of $D\left(P_{X;\theta_0}\middle\|P_{X;\hat{\theta}(X^{N_0})}\right)$ at $\theta_0$, we can get

$$D\left(P_{X;\theta_0}\middle\|P_{X;\hat{\theta}(X^{N_0})}\right)$$

$$= \sum_{x\in X} P_{X;\theta_0}(x)\log\frac{P_{X;\theta_0}(x)}{P_{X;\hat{\theta}(X^{N_0})}(x)}$$

$$= -\sum_{x\in X} P_{X;\theta_0}(x)\log\frac{P_{X;\hat{\theta}(X^{N_0})}(x)}{P_{X;\theta_0}(x)}$$

$$= -\sum_{x\in X} P_{X;\theta_0}(x)\log\left(1 + \frac{P_{X;\hat{\theta}(X^{N_0})}(x) - P_{X;\theta_0}(x)}{P_{X;\theta_0}(x)}\right)$$

$$= -\sum_{x\in X} P_{X;\theta_0}(x)\left(\left(\frac{P_{X;\hat{\theta}(X^{N_0})}(x) - P_{X;\theta_0}(x)}{P_{X;\theta_0}(x)}\right) - \frac{1}{2}\left(\frac{P_{X;\hat{\theta}(X^{N_0})}(x) - P_{X;\theta_0}(x)}{P_{X;\theta_0}(x)}\right)^2\right) + o(|\hat{\theta}(X^{N_0}) - \theta_0|^2)$$

$$= \frac{1}{2}\sum_{x\in X}\frac{\left(P_{X;\hat{\theta}(X^{N_0})}(x) - P_{X;\theta_0}(x)\right)^2}{P_{X;\theta_0}(x)} + o(|\hat{\theta}(X^{N_0}) - \theta_0|^2) \tag{17}$$

We denote $\delta$ as a small constant, and we can rewrite (6) as

$$\mathbb{E}\left[D\left(P_{X;\theta_0}\middle\|P_{X;\hat{\theta}(X^{N_0})}\right)\right]$$

$$= \sum_{X^{N_0}} P_{X^n;\theta_0}(X^{N_0})D\left(P_{X;\theta_0}\middle\|P_{X;\hat{\theta}(X^{N_0})}\right)$$

$$= \sum_{X^{N_0}} P_{X^n;\theta_0}(X^{N_0})\left(\frac{1}{2}\sum_{x\in X}\frac{\left(P_{X;\hat{\theta}(X^{N_0})}(x) - P_{X;\theta_0}(x)\right)^2}{P_{X;\theta_0}(x)} + o(|\hat{\theta}(X^{N_0}) - \theta_0|^2)\right)$$

$$= \sum_{X^{N_0}} P_{X^n;\theta_0}(X^{N_0})\left(\frac{1}{2}\sum_{x\in X}\frac{\left(\frac{\partial P_{X;\theta_0}(x)}{\partial\theta_0}(\hat{\theta}(X^{N_0}) - \theta_0)\right)^2}{P_{X;\theta_0}(x)} + o(|\hat{\theta}(X^{N_0}) - \theta_0|^2)\right)$$

$$= \sum_{\{X^{N_0}:|\hat{\theta}(X^{N_0})-\theta_0|^2<\delta\}} P_{X^n;\theta_0}(X^{N_0})\left(\frac{1}{2}\sum_{x\in X}\frac{\left(\frac{\partial P_{X;\theta_0}(x)}{\partial\theta_0}(\hat{\theta}(X^{N_0}) - \theta_0)\right)^2}{P_{X;\theta_0}(x)} + o(|\hat{\theta}(X^{N_0}) - \theta_0|^2)\right)$$

$$+ \sum_{\{X^{N_0}:|\hat{\theta}(X^{N_0})-\theta_0|^2\geq\delta\}} P_{X^n;\theta_0}(X^{N_0})\left(\frac{1}{2}\sum_{x\in X}\frac{\left(\frac{\partial P_{X;\theta_0}(x)}{\partial\theta_0}(\hat{\theta}(X^{N_0}) - \theta_0)\right)^2}{P_{X;\theta_0}(x)} + o(|\hat{\theta}(X^{N_0}) - \theta_0|^2)\right) \tag{18}$$

To facilitate the subsequent proof, we introduce the concept of "Dot Equal".

**Definition 9.** *(Dot Equal ($\doteq$)) Specifically, given two functions $f(n)$ and $g(n)$, the notation $f(n)\doteq g(n)$ is defined as*

$$f(n)\doteq g(n) \quad \Leftrightarrow \quad \lim_{n\to\infty} \frac{1}{n} \log \frac{f(n)}{g(n)} = 0, \tag{19}$$

*which shows that $f(n)$ and $g(n)$ have the same exponential decaying rate.*

We denote $\hat{P}_{X^{N_0}}$ as the empirical distribution of $X^{N_0}$. Applying Sanov's Theorem to (18), we can know that

$$P_{X^n;\theta_0}(X^{N_0}) \doteq e^{-N_0 D\left(\hat{P}_{X^{N_0}} \| P_{X;\theta_0}\right)} \tag{20}$$

Then, we aim to establish a connection between (20) and $|\hat{\theta}(X^{N_0}) - \theta_0|^2$. From (17), we can know that the $D\left(\hat{P}_{X^{N_0}} \| P_{X;\theta_0}\right)$ in (20) can be transformed to

$$D\left(\hat{P}_{X^{N_0}} \| P_{X;\theta_0}\right) = \frac{1}{2} \sum_{x\in X} \frac{\left(\hat{P}_{X^{N_0}}(x) - P_{X;\theta_0}(x)\right)^2}{\hat{P}_{X^{N_0}}(x)} + o(|\hat{\theta}(X^{N_0}) - \theta_0|^2)$$

$$= \frac{1}{2} \sum_{x\in X} \frac{\left(\hat{P}_{X^{N_0}}(x) - P_{X;\theta_0}(x)\right)^2}{P_{X;\theta_0}(x)} + o(|\hat{\theta}(X^{N_0}) - \theta_0|^2) \tag{21}$$

From the characteristics of MLE, we can know that

$$\mathbb{E}_{\hat{P}_{X^{N_0}}}\left[\frac{\partial \log P_{X;\hat{\theta}(X^{N_0})}(x)}{\partial \hat{\theta}}\right]$$

$$= 0$$

$$= \mathbb{E}_{\hat{P}_{X^{N_0}}}\left[\frac{\partial \log P_{X;\theta_0}(x)}{\partial \theta_0}\right] + \mathbb{E}_{\hat{P}_{X^{N_0}}}\left[\frac{\partial^2 \log P_{X;\theta_0}(x)}{\partial \theta_0^2}\right] \left(\hat{\theta}(X^{N_0}) - \theta_0\right) + O(|\hat{\theta}(X^{N_0}) - \theta_0|^2), \tag{22}$$

which can be transform to

$$\left(\hat{\theta}(X^{N_0}) - \theta_0\right) + O(|\hat{\theta}(X^{N_0}) - \theta_0|^2)$$

$$= -\frac{\mathbb{E}_{\hat{P}_{X^{N_0}}}\left[\frac{\partial \log P_{X;\theta_0}(x)}{\partial \theta_0}\right]}{\mathbb{E}_{\hat{P}_{X^{N_0}}}\left[\frac{\partial^2 \log P_{X;\theta_0}(x)}{\partial \theta_0^2}\right]}$$

$$= -\frac{\mathbb{E}_{\hat{P}_{X^{N_0}}}\left[\frac{\frac{\partial P_{X;\theta_0}(x)}{\partial \theta_0}}{P_{X;\theta_0}(x)}\right]}{\mathbb{E}_{\hat{P}_{X^{N_0}}}\left[\frac{\partial^2 \log P_{X;\theta_0}(x)}{\partial \theta_0^2}\right]}$$

$$= \frac{\sum_{x\in\mathcal{X}} \left(\hat{P}_{X^{N_0}}(x) - P_{X;\theta_0}(x)\right) \frac{\frac{\partial P_{X;\theta_0}(x)}{\partial \theta_0}}{P_{X;\theta_0}(x)}}{J(\theta_0)} \tag{23}$$

Using the Cauchy-Schwarz inequality, we can obtain

$$\sum_{x\in X} \frac{\left(\hat{P}_{X^{N_0}}(x) - P_{X;\theta_0}(x)\right)^2}{P_{X;\theta_0}(x)} \cdot \sum_{x} \frac{(\frac{\partial P_{X;\theta_0}(x)}{\partial \theta_0})^2}{P_{X;\theta_0}(x)} \geq \left(\sum_{x\in\mathcal{X}} \left(\hat{P}_{X^{N_0}}(x) - P_{X;\theta_0}(x)\right) \frac{\frac{\partial P_{X;\theta_0}(x)}{\partial \theta_0}}{P_{X;\theta_0}(x)}\right)^2 \tag{24}$$

where

$$\sum_x \frac{(\frac{\partial P_{X;\theta_0}(x)}{\partial \theta_0})^2}{P_{X;\theta_0}(x)} = \sum_x P_{X;\theta_0}(x) \left( \frac{1}{P_{X;\theta_0}(x)} \frac{\partial P_{X;\theta_0}(x)}{\partial \theta_0} \right)^2 = \sum_x P_{X;\theta_0}(x) \left( \frac{\partial \log P_{X;\theta_0}(x)}{\partial \theta_0} \right)^2 = J(\theta_0)$$
(25)

Combining with (21), (23), and (25), the inequality (24) can be transformed to

$$D\left(\hat{P}_{X^{N_0}} \middle\| P_{X;\theta_0}\right)$$
$$= \frac{1}{2} \sum_{x \in X} \frac{\left(\hat{P}_{X^{N_0}}(x) - P_{X;\theta_0}(x)\right)^2}{P_{X;\theta_0}(x)} + o(|\hat{\theta}(X^{N_0}) - \theta_0|^2)$$
$$\geq \frac{1}{2} J(\theta_0) \left(\hat{\theta}(X^{N_0}) - \theta_0 + O(|\hat{\theta}(X^{N_0}) - \theta_0|^2)\right)^2 + o(|\hat{\theta}(X^{N_0}) - \theta_0|^2)$$
$$= \frac{1}{2} J(\theta_0) |\hat{\theta}(X^{N_0}) - \theta_0|^2 + o(|\hat{\theta}(X^{N_0}) - \theta_0|^2)$$
(26)

Combining (20) and (26), we can know that

$$P_{X^n;\theta_0}(X^{N_0}) \doteq e^{-N_0 D\left(\hat{P}_{X^{N_0}} \middle\| P_{X;\theta_0}\right)} \leq e^{\frac{-N_0 J(\theta_0)|\hat{\theta}(X^{N_0}) - \theta_0|^2}{2}}$$
(27)

For the first term in (18), the $\|\hat{\theta}(X^{N_0}) - \theta_0\|$ is small enough for us to omit the term $o(|\hat{\theta}(X^{N_0}) - \theta_0|^2)$. As for the second term in in (18), even though the magnitude of $\|\hat{\theta}(X^{N_0}) - \theta_0\|$ is no longer negligible, the probability of such sequences is $O(e^{\frac{-N_0 J(\theta_0)|\hat{\theta}(X^{N_0}) - \theta_0|^2}{2}})$ by (27), which is exponentially decaying with $N_0$ such that the second term is $o(\frac{1}{N_0})$. By transfering (18), we can get

$$\mathbb{E}\left[D\left(P_{X;\theta_0} \middle\| P_{X;\hat{\theta}(X^{N_0})}\right)\right]$$
$$= \sum_{\left\{X^{N_0}:|\hat{\theta}(X^{N_0}) - \theta_0|^2 < \delta\right\}} P_{X^n;\theta_0}(X^{N_0}) \left(\frac{1}{2} \sum_{x \in X} \frac{\left(\frac{\partial P_{X;\theta_0}(x)}{\partial \theta_0}(\hat{\theta}(X^{N_0}) - \theta_0)\right)^2}{P_{X;\theta_0}(x)}\right) + o(|\hat{\theta}(X^{N_0}) - \theta_0|^2)$$
$$+ \sum_{\left\{X^{N_0}:|\hat{\theta}(X^{N_0}) - \theta_0|^2 \geq \delta\right\}} P_{X^n;\theta_0}(X^{N_0}) \left(\frac{1}{2} \sum_{x \in X} \frac{\left(\frac{\partial P_{X;\theta_0}(x)}{\partial \theta_0}(\hat{\theta}(X^{N_0}) - \theta_0)\right)^2}{P_{X;\theta_0}(x)}\right) + o(\frac{1}{N_0})$$
$$= \frac{1}{2}\mathbb{E}\left[\sum_{x \in X} \frac{\left(\frac{\partial P_{X;\theta_0}(x)}{\partial \theta_0}(\hat{\theta}(X^{N_0}) - \theta_0)\right)^2}{P_{X;\theta_0}(x)}\right] + o(\frac{1}{N_0}).$$
(28)

We then transform (28) with (25)

$$\frac{1}{2}\mathbb{E}\left[\sum_{x \in X} \frac{\left(\frac{\partial P_{X;\theta_0}(x)}{\partial \theta_0}(\hat{\theta}(X^{N_0}) - \theta_0)\right)^2}{P_{X;\theta_0}(x)}\right]$$
$$= \frac{1}{2}\mathbb{E}\left[\left(\hat{\theta} - \theta_0\right)^2\right] \sum_x \frac{(\frac{\partial P_{X;\theta_0}(x)}{\partial \theta_0})^2}{P_{X;\theta_0}(x)}$$
$$= \frac{1}{2}\mathbb{E}\left[\left(\hat{\theta} - \theta_0\right)^2\right] J(\theta_0)$$
(29)

Combining (28), (29), we can get

$$\mathbb{E}\left[D\left(P_{X;\theta_0} \middle\| P_{X;\hat{\theta}}\right)\right] = \frac{1}{2}\mathbb{E}\left[\left(\hat{\theta} - \theta_0\right)^2\right] J(\theta_0) + o(\frac{1}{N_0}).$$
(30)

$\square$

From (16), we can establish the relationship between the proposed proposed measure (6) and the mean squared error. Then, by using the (2), the K-L measure is

$$\frac{1}{2N_0} + o(\frac{1}{N_0}). \tag{31}$$

## C.2 Proof of Theorem 4

In this theorem, we are using samples from both target task and source task for our maximum likelihood estimation, so our optimization problem becomes

$$\hat{\theta} = \arg\max_\theta L_n(\theta), \tag{32}$$

where, when using $N_0$ samples from target task, and $n_1$ samples from source task

$$L_n(\theta) \triangleq \frac{1}{N_0 + n_1} \sum_{x \in X^{N_0}} \log P_{X;\theta}(x) + \frac{1}{N_0 + n_1} \sum_{x \in X^{n_1}} \log P_{X;\theta}(x). \tag{33}$$

And, we also define the expectation of our estimator, which is somewhere between $\theta_0$ and $\theta_1$ to be $E_{\hat{\theta}}$

$$E_{\hat{\theta}} = \arg\max_\theta L(\theta), \tag{34}$$

where $L(\theta)$ denotes the expectation of $L_n(\theta)$

$$L(\theta) \triangleq \frac{N_0}{N_0 + n_1} \mathbb{E}_{P_{X;\theta_0}} \left[ \log P_{X;\theta}(x) \right] + \frac{n_1}{N_0 + n_1} \mathbb{E}_{P_{X;\theta_1}} \left[ \log P_{X;\theta}(x) \right]. \tag{35}$$

We could equivalently transform (34) into

$$E_{\hat{\theta}} = \arg\min_\theta \frac{N_0 D \left( P_{X;\theta_0} \| P_{X;\theta} \right)}{N_0 + n_1} + \frac{n_1 D \left( P_{X;\theta_1} \| P_{X;\theta} \right)}{N_0 + n_1} \tag{36}$$

**Lemma 10.** *By taking argmin of* (36) *we can get*

$$P_{X;E_{\hat{\theta}}} = \frac{N_0 P_{X;\theta_0} + n_1 P_{X;\theta_1}}{N_0 + n_1}, \tag{37}$$

*By doing a Taylor Expansion of* (37) *around $\theta'$, which is in the neighbourhood of $\theta_1, \theta_2$ and $E_{\hat{\theta}}$, we can get*

$$E_{\hat{\theta}} = \frac{N_0 \theta_0 + n_1 \theta_1}{N_0 + n_1} + O\left(\frac{1}{N_0 + n_1}\right), \tag{38}$$

*Proof.* From (36), we know that

$$E_{\hat{\theta}} = \arg\min_\theta \frac{N_0 D \left( P_{X;\theta_0} \| P_{X;\theta} \right)}{N_0 + n_1} + \frac{n_1 D \left( P_{X;\theta_1} \| P_{X;\theta} \right)}{N_0 + n_1}$$

$$= \arg\min_\theta \frac{N_0}{N_0 + n_1} \sum_{x \in \mathcal{X}} P_{X;\theta_0}(x) \log \frac{P_{X;\theta_0}(x)}{P_{X;\theta}(x)} + \frac{n_1}{N_0 + n_1} \sum_{x \in \mathcal{X}} P_{X;\theta_1}(x) \log \frac{P_{X;\theta_1}(x)}{P_{X;\theta}(x)} \tag{39}$$

To minimize the weighted K-L divergence. We treat $P_{X;\theta}(x), \forall x \in \mathcal{X}$ as the variable with the constraint $\sum_x P_{X;\theta}(x) = 1$, then we form the Lagrangian:

$$\text{Lagrangian}(P, \lambda) = -\sum_x \left( \frac{N_0}{N_0 + n_1} P_{X;\theta_0}(x) + \frac{n_1}{N_0 + n_1} P_{X;\theta_1}(x) \right) \log P_{X;\theta}(x) + \lambda \left( \sum_x P_{X;\theta}(x) - 1 \right) \tag{40}$$

Taking the derivative with respect to $P_{X;\theta}(x)$ and setting it to zero gives:

$$\frac{\partial \text{Lagrangian}(P, \lambda)}{\partial P_{X;\theta}(x)} = -\frac{\frac{N_0}{N_0+n_1} P_{X;\theta_0}(x) + \frac{n_1}{N_0+n_1} P_{X;\theta_1}(x)}{P_{X;\theta}(x)} + \lambda = 0. \tag{41}$$

So

$$P_{X;\theta}(x) = \frac{\frac{N_0}{N_0+n_1}P_{X;\theta_0}(x) + \frac{n_1}{N_0+n_1}P_{X;\theta_1}(x)}{\lambda}. \tag{42}$$

Normalizing $P_{X;\theta}(x)$ gives $\lambda = 1$, hence the optimal solution is:

$$P_{X;E_{\hat{\theta}}}(x) = \frac{N_0}{N_0+n_1}P_{X;\theta_0}(x) + \frac{n_1}{N_0+n_1}P_{X;\theta_1}(x), \forall x \in \mathcal{X}, \tag{43}$$

which corresponds to (37).

Then we begin to prove (38). By doing a Taylor Expansion of (43) around $\theta_0$, we can get

$$P_{X;\theta_0}(x) + \frac{\partial P_{X;\theta_0}(x)}{\partial \theta}(E_{\hat{\theta}} - \theta_0) + O(|E_{\hat{\theta}} - \theta_0|^2)$$
$$= \frac{N_0}{N_0+n_1}P_{X;\theta_0}(x) + \frac{n_1}{N_0+n_1}\left(P_{X;\theta_0}(x) + \frac{\partial P_{X;\theta_0}(x)}{\partial \theta}(\theta_1 - \theta_0) + O(|\theta_0 - \theta_1|^2)\right) \tag{44}$$

From (44) we can get

$$E_{\hat{\theta}} = \frac{N_0\theta_0 + n_1\theta_1}{N_0+n_1} + O(|\theta_0 - \theta_1|^2), \tag{45}$$

So we can get

$$E_{\hat{\theta}} = \frac{N_0\theta_0 + n_1\theta_1}{N_0+n_1} + O\left(\frac{1}{N_0+n_1}\right). \tag{46}$$

$$\square$$

**Lemma 11.** *We assume that the following regularity conditions hold:*

*1. The log-likelihood function is twice continuously differentiable in the neighborhood of $\theta_0$.*

*2. The Fisher information $J(\theta_0)$ is positive and finite.*

*Then, the estimator $\hat{\theta}$ is asymptotically normal,i.e.,*

$$\sqrt{N_0+n_1}(\hat{\theta} - E_{\hat{\theta}}) \xrightarrow{d} \mathcal{N}\left(0, \frac{1}{J(\theta_0)}\right). \tag{47}$$

*Proof.* Since $\hat{\theta}$ is a maximizer of $L_n(\theta)$, $L'_n(\hat{\theta}) = 0 = L'_n(E_{\hat{\theta}}) + L''_n(E_{\hat{\theta}})(\hat{\theta} - E_{\hat{\theta}}) + O\left(\frac{1}{N_0}\right)$. Therefore,

$$\sqrt{N_0+n_1}(\hat{\theta} - E_{\hat{\theta}}) = -\frac{\sqrt{N_0+n_1}L'_n(E_{\hat{\theta}})}{L''_n(E_{\hat{\theta}})} + O\left(\frac{1}{\sqrt{N_0}}\right). \tag{48}$$

Since $E_{\hat{\theta}}$ maximizes $L(\theta)$, $L'(E_{\hat{\theta}}) = \frac{N_0}{N_0+n_1}\mathbb{E}_{\theta_0}\left[\frac{\partial \log P_{X;E_{\hat{\theta}}}(x)}{\partial x}\right] + \frac{n_1}{N_0+n_1}\mathbb{E}_{\theta_1}\left[\frac{\partial \log P_{X;E_{\hat{\theta}}}(x_i)}{\partial x}\right] = 0$.
Therefore,

$$\sqrt{N_0 + n_1} L'_n(E_{\hat{\theta}})$$

$$= \sqrt{\frac{N_0}{N_0+n_1}}\left(\sqrt{\frac{1}{N_0}}\sum_{x \in X^{N_0}}\frac{\partial \log P_{X;E_{\hat{\theta}}}(x)}{\partial \theta}\right) + \sqrt{\frac{n_1}{N_0+n_1}}\left(\sqrt{\frac{1}{n_1}}\sum_{x \in X^{n_1}}\frac{\partial \log P_{X;E_{\hat{\theta}}}(x)}{\partial \theta}\right)$$

$$= \sqrt{\frac{N_0}{N_0+n_1}}\left(\sqrt{\frac{1}{N_0}}\sum_{x \in X^{N_0}}\frac{\partial \log P_{X;E_{\hat{\theta}}}(x)}{\partial \theta}\right) + \sqrt{\frac{n_1}{N_0+n_1}}\left(\sqrt{\frac{1}{n_1}}\sum_{x \in X^{n_1}}\frac{\partial \log P_{X;E_{\hat{\theta}}}(x)}{\partial \theta}\right)$$

$$- \frac{N_0}{\sqrt{N_0+n_1}}\mathbb{E}_{\theta_0}\left[\frac{\partial \log P_{X;E_{\hat{\theta}}}(x)}{\partial x}\right] - \frac{n_1}{\sqrt{N_0+n_1}}\mathbb{E}_{\theta_1}\left[\frac{\partial \log P_{X;E_{\hat{\theta}}}(x)}{\partial x}\right]$$

$$= \sqrt{\frac{N_0}{N_0+n_1}}\left(\sqrt{\frac{1}{N_0}}\sum_{x \in X^{N_0}}\frac{\partial \log P_{X;E_{\hat{\theta}}}(x)}{\partial \theta} - \sqrt{N_0}\mathbb{E}_{\theta_0}\left[\frac{\partial \log P_{X;E_{\hat{\theta}}}(x)}{\partial \theta}\right]\right)$$

$$+ \sqrt{\frac{n_1}{N_0+n_1}}\left(\sqrt{\frac{1}{n_1}}\sum_{x \in X^{n_1}}\frac{\partial \log P_{X;E_{\hat{\theta}}}(x)}{\partial \theta} - \sqrt{n_1}\mathbb{E}_{\theta_1}\left[\frac{\partial \log P_{X;E_{\hat{\theta}}}(x)}{\partial \theta}\right]\right) \tag{49}$$

Applying the Central Limit Theorem to (49), we can get

$$\sqrt{N_0 + n_1} L'_n(E_{\hat{\theta}}) \xrightarrow{a.s.} \mathcal{N}\left(0, \frac{N_0}{N_0+n_1}\left(\mathbb{E}_{\theta_0}\left[\left(\frac{\partial \log P_{X;E_{\hat{\theta}}}(x)}{\partial \theta}\right)^2\right] - \mathbb{E}_{\theta_0}\left[\left(\frac{\partial \log P_{X;E_{\hat{\theta}}}(x)}{\partial \theta}\right)\right]^2\right) + \right.$$

$$\left. \frac{n_1}{N_0+n_1}\left(\mathbb{E}_{\theta_1}\left[\left(\frac{\partial \log P_{X;E_{\hat{\theta}}}(x)}{\partial \theta}\right)^2\right] - \mathbb{E}_{\theta_1}\left[\left(\frac{\partial \log P_{X;E_{\hat{\theta}}}(x)}{\partial \theta}\right)\right]^2\right)\right) \tag{50}$$

By taking Taylor expansion of $E_{\hat{\theta}}$ at $\theta_0$, we can get

$$\mathbb{E}_{\theta_0}\left[\left(\frac{\partial \log P_{X;E_{\hat{\theta}}}(x)}{\partial \theta}\right)^2\right]$$

$$= \mathbb{E}_{\theta_0}\left[\left(\frac{\partial \log P_{X;\theta_0}(x)}{\partial \theta} + \frac{\partial}{\partial \theta}\frac{\partial \log P_{X;\theta_0}(x)}{\partial \theta}(E_{\hat{\theta}} - \theta_0) + O(\frac{1}{N_0+n_1})\right)^2\right]$$

$$= J(\theta_0) + (E_{\hat{\theta}} - \theta_0)\mathbb{E}_{\theta_0}\left[\frac{\partial \log P_{X;E_{\hat{\theta}}}(x)}{\partial \theta}\frac{\partial^2 \log P_{X;E_{\hat{\theta}}}(x)}{\partial \theta^2}\right] + O(\frac{1}{N_0+n_1})$$

$$= J(\theta_0) + O(\frac{1}{\sqrt{N_0+n_1}}) \tag{51}$$

and

$$\mathbb{E}_{\theta_0}\left[\left(\frac{\partial \log P_{X;E_{\hat{\theta}}}(x)}{\partial \theta}\right)\right]^2$$

$$= \mathbb{E}_{\theta_0}\left[\left(\frac{\partial \log P_{X;\theta_0}(x)}{\partial \theta} + \frac{\partial}{\partial \theta}\frac{\partial \log P_{X;\theta_0}(x)}{\partial \theta}(E_{\hat{\theta}} - \theta_0) + O(\frac{1}{N_0+n_1})\right)\right]^2$$

$$= \mathbb{E}_{\theta_0}\left[\left(\frac{\partial}{\partial \theta}\frac{\partial \log P_{X;\theta_0}(x)}{\partial \theta}(E_{\hat{\theta}} - \theta_0) + O(\frac{1}{N_0+n_1})\right)\right]^2$$

$$= (E_{\hat{\theta}} - \theta_0)^2 \mathbb{E}_{\theta_0}\left[\left(\frac{\partial}{\partial \theta}\frac{\partial \log P_{X;\theta_0}(x)}{\partial \theta}\right)\right]^2 + o(\frac{1}{N_0+n_1})$$

$$= O(\frac{1}{N_0+n_1}) \tag{52}$$

By combining (50), (51), and (52), we can get

$$\text{var}\left(\sqrt{N_0 + n_1}L'_n(E_{\hat{\theta}})\right) = \frac{N_0}{N_0 + n_1}J(\theta_0) + \frac{n_1}{N_0 + n_1}J(\theta_1) + O(\frac{1}{\sqrt{N_0 + n_1}}) \tag{53}$$

Additionally, we know $L''_n(E_{\hat{\theta}}) \xrightarrow{p} -J(E_{\hat{\theta}})$. Combining with (48)(53), we know that

$$\sqrt{N_0 + n_1}(\hat{\theta} - E_{\hat{\theta}}) \xrightarrow{d} \mathcal{N}\left(0, \frac{\frac{N_0}{N_0+n_1}J(\theta_0) + \frac{n_1}{N_0+n_1}J(\theta_1)}{J^2(E_{\hat{\theta}})}\right) \tag{54}$$

Under the assumption that $\theta_0, \theta_1, E_{\hat{\theta}}$ are sufficiently close to each other, we can easily deduce that the difference among $J(\theta_0), J(\theta_1)$ and $J(E_{\hat{\theta}})$ is $O(\frac{1}{\sqrt{N_0+n_1}})$. We can easily get

$$\sqrt{N_0 + n_1}(\hat{\theta} - E_{\hat{\theta}}) \xrightarrow{d} \mathcal{N}\left(0, \frac{1}{J(\theta_0)}\right) \tag{55}$$

$\square$

Therefore, the limit of $\mathbb{E}\left[\left(\hat{\theta} - E_{\hat{\theta}}\right)^2\right]$ is

$$\frac{1}{(N_0 + n_1)J(\theta_0)} \tag{56}$$

Combining (6) and (29), we know that

$$
\begin{aligned}
&\mathbb{E}\left[D\left(P_{X;\theta_0}\|P_{X;\hat{\theta}}\right)\right] \\
&= \frac{1}{2}J(\theta_0)\mathbb{E}\left[\left(\hat{\theta} - \theta_0\right)^2\right] + o(\frac{1}{N_0}) \\
&= \frac{1}{2}J(\theta_0)\left(\mathbb{E}\left[\left(\hat{\theta} - E_{\hat{\theta}}\right)^2\right] + \mathbb{E}\left[2\left(\hat{\theta} - E_{\hat{\theta}}\right)\left(E_{\hat{\theta}} - \theta_0\right)\right] + \mathbb{E}\left[\left(E_{\hat{\theta}} - \theta_0\right)^2\right]\right) + o(\frac{1}{N_0}) \\
&= \frac{1}{2}J(\theta_0)\left(\mathbb{E}\left[\left(\hat{\theta} - E_{\hat{\theta}}\right)^2\right] + \mathbb{E}\left[\left(E_{\hat{\theta}} - \theta_0\right)^2\right]\right) + o(\frac{1}{N_0}) \tag{57}
\end{aligned}
$$

Combining (57), (38) and (56), we know that the proposed measure is

$$\frac{1}{2}\frac{1}{N_0 + n_1} + \frac{1}{2}J(\theta_0)\frac{n_1^2}{(N_0 + n_1)^2}\left(\theta_1 - \theta_0\right)^2 + o(\frac{1}{N_0}) \tag{58}$$

### C.3  Proof of Proposition 5

Similar to (16), we can get

$$
\begin{aligned}
&\mathbb{E}\left[D\left(P_{X;\underline{\theta}_0}\|P_{X;\underline{\hat{\theta}}}\right)\right] \\
&= \frac{1}{2}tr\left(J(\underline{\theta}_0)\mathbb{E}\left[\left(\underline{\hat{\theta}} - \underline{\theta}_0\right)\left(\underline{\hat{\theta}} - \underline{\theta}_0\right)^T\right]\right) + o(\frac{1}{N_0}) \tag{59}
\end{aligned}
$$

Combining with Lemma 1, we will know that the K-L measure is

$$\frac{1}{2}tr\left(J(\underline{\theta}_0)\left(J(\underline{\theta}_0)^{-1}\frac{1}{N_0} + o(\frac{1}{N_0})\right)\right) = \frac{d}{2N_0} + o(\frac{1}{N_0}) \tag{60}$$

## C.4 Proof of Proposition 6

Similar to (57), we can get

$$
\mathbb{E}\left[ D\left( P_{X;\underline{\theta}_0} \middle\| P_{X;\underline{\hat{\theta}}} \right) \right]
$$
$$
= \frac{1}{2} tr\left( J(\underline{\theta}_0)\mathbb{E}\left[ \left( \underline{\hat{\theta}} - \underline{\theta}_0 \right)\left( \underline{\hat{\theta}} - \underline{\theta}_0 \right)^T \right] \right) + o(\frac{1}{N_0})
$$
$$
= \frac{1}{2}\left( tr\left( J(\underline{\theta}_0)\mathbb{E}\left[ \left( \underline{\hat{\theta}} - E_{\underline{\hat{\theta}}} \right)\left( \underline{\hat{\theta}} - E_{\underline{\hat{\theta}}} \right)^T \right] \right) + tr\left( J(\underline{\theta}_0)\mathbb{E}\left[ \left( E_{\underline{\hat{\theta}}} - \underline{\theta}_0 \right)\left( E_{\underline{\hat{\theta}}} - \underline{\theta}_0 \right)^T \right] \right) \right) + o(\frac{1}{N_0})
$$
$$
\tag{61}
$$

Similar to (47), we can get

$$
\sqrt{N_0 + n_1}\left( \underline{\hat{\theta}} - E_{\underline{\hat{\theta}}} \right) \xrightarrow{d} \mathcal{N}\left( 0, J(\underline{\theta}_0)^{-1} \right) \tag{62}
$$

So we can know that $\mathbb{E}\left[ \left( \underline{\hat{\theta}} - E_{\underline{\hat{\theta}}} \right)^2 \right]$ has the limit

$$
\frac{1}{(N_0 + n_1)} J(\underline{\theta}_0)^{-1} \tag{63}
$$

The same to (38), we can get

$$
E_{\underline{\hat{\theta}}} = \frac{N_0\underline{\theta}_0 + n_1\underline{\theta}_1}{N_0 + n_1} + O\left( \frac{1}{N_0 + n_1} \right), \tag{64}
$$

Combining (61), (63) and (64), we can know that the K-L measure is

$$
\frac{d}{2}\left( \frac{1}{N_0 + n_1} + \frac{n_1^2}{(N_0 + n_1)^2}t \right) + o\left( \frac{1}{N_0 + n_1} \right), \tag{65}
$$

where we denote

$$
t \triangleq \frac{(\underline{\theta}_1 - \underline{\theta}_0)^T J(\underline{\theta}_0)(\underline{\theta}_1 - \underline{\theta}_0)}{d}. \tag{66}
$$

## C.5 Proof of Theorem 7

*Proof.* Similar to (47), we can get

$$
\sqrt{N_0 + \sum_{i=1}^{K} n_i}\left( \underline{\hat{\theta}} - E_{\underline{\hat{\theta}}} \right) \xrightarrow{d} \mathcal{N}\left( 0, J(\underline{\theta}_0)^{-1} \right) \tag{67}
$$

So we can know that $\mathbb{E}\left[ \left( \underline{\hat{\theta}} - E_{\underline{\hat{\theta}}} \right)^2 \right]$ has the limit

$$
\frac{1}{(N_0 + s)} J(\underline{\theta}_0)^{-1} \tag{68}
$$

Similar to (38), we can get

$$
E_{\underline{\hat{\theta}}} = \frac{N_0\underline{\theta}_0 + \sum_{i=1}^{k} n_i\underline{\theta}_i}{N_0 + \sum_{i=1}^{K} n_i} + O\left( \frac{1}{N_0 + \sum_{i=1}^{K} n_i} \right) = \frac{N_0\underline{\theta}_0 + \sum_{i=1}^{k} n_i\underline{\theta}_i}{N_0 + s} + O\left( \frac{1}{N_0 + s} \right) \tag{69}
$$

Combining (61), (68) and (69), for the d-demension parameter, we can know that the K-L measure is

$$
\frac{d}{2}\left( \frac{1}{N_0 + s} + \frac{s^2}{(N_0 + s)^2 d}tr\left( \mathbf{J}(\underline{\theta}_0)\left( \sum_{i=1}^{k} \alpha_i(\underline{\theta} - \underline{\theta}_0) \right)\left( \sum_{i=1}^{k} \alpha_i(\underline{\theta}_i - \underline{\theta}_0) \right)^T \right) \right)
$$
$$
= \frac{d}{2}\left( \frac{1}{N_0 + s} + \frac{s^2}{(N_0 + s)^2}\frac{\underline{\alpha}^T\Theta^T J(\underline{\theta}_0)\Theta\underline{\alpha}}{d} \right) + o\left( \frac{1}{N_0 + s} \right)
$$
$$
\tag{70}
$$

where $\underline{\alpha} = [\alpha_1, \ldots, \alpha_K]^T$ is a K-dimensional vector, and

$$\Theta^{d \times K} = [\underline{\theta}_1 - \underline{\theta}_0, \ldots, \underline{\theta}_K - \underline{\theta}_0].\tag{71}$$

$\square$

# D   Experiment Details

## D.1   Details information of LoRA framework experiments.

Table 6: Multi-Source Transfer with LoRA on Office-Home. We apply LoRA on ViT-B backbone for PEFT.

| Method | Backbone | Office-Home | | | | |
|---|---|---|---|---|---|---|
| | | →Ar | →Cl | →Pr | →Rw | Avg |
| *Supervised-10-shots Source-Ablation:* | | | | | | |
| Target-Only | ViT-B | 59.8 | 42.2 | 69.5 | 72.0 | 60.9 |
| Single-Source-avg | ViT-B | 72.2 | 59.9 | 82.6 | 81.0 | 73.9 |
| Single-Source-best | ViT-B | 74.4 | 61.8 | 84.9 | 81.9 | 75.8 |
| AllSources ∪ Target | ViT-B | 81.1 | 66.0 | 88.0 | 89.2 | 81.1 |
| OTQMS (Ours) | ViT-B | **81.5** | **68.0** | **89.2** | **90.3** | **82.3** |

## D.2   Experimental Design and Model Adaptation

To ensure consistency in the experimental setup, we first evaluate the performance of different methods on the DomainNet and Office-Home datasets by adapting their settings to align with ours, such as the backbone, dataset, and early stopping criteria. Specifically, for the MADA method, we adjusted the preset of keeping 5% of labeled target samples to 10-shots per class target samples while maintaining other conditions. And in turn, ours is adapted to the WADN settings, equipped with a 3-layer ConvNet and evaluated on Digits dataset.

Table 7: **Performance on Digits Dataset.** The arrows indicate transferring from the rest tasks. "3Conv" denotes the backbone with 3 convolution layers.

| Method | Backbone | Digits | | | | |
|---|---|---|---|---|---|---|
| | | →mt | →sv | →sy | →up | Avg |
| *Following settings of WADN:* | | | | | | |
| WADN[22] | 3Conv | 88.3 | **70.6** | 81.5 | **90.5** | 82.7 |
| AllSources ∪ Target | 3Conv | 92.6 | 67.1 | 82.5 | 88.8 | 82.8 |
| OTQMS (Ours) | 3Conv | **93.8** | 67.1 | **83.3** | 89.1 | **83.3** |

### D.3 Single-Source transfer performance details.

Table 8: **Single-Source Transfer Performance on DomainNet.** The details accuracy information of the "Single-Source-∗" lines of Table 2.

| Target Domain | Backbone | Source Domain | | | | | | | |
|---|---|---|---|---|---|---|---|---|---|
| | | Clipart | Infograph | Painting | Quickdraw | Real | Sketch | Avg | Best |
| Clipart | ViT-S | - | 46.5 | 55.4 | 30.3 | 60.2 | 59.7 | 50.4 | 60.2 |
| Infograph | ViT-S | 25.6 | - | 25.3 | 7.3 | 28.0 | 24.4 | 22.1 | 28.0 |
| Painting | ViT-S | 49.6 | 47.3 | - | 22.4 | 55.4 | 49.6 | 44.9 | 55.4 |
| Quickdraw | ViT-S | 26.9 | 18.1 | 23.9 | - | 25.9 | 28.4 | 24.7 | 28.4 |
| Real | ViT-S | 64.6 | 62.2 | 66.0 | 38.5 | - | 62.7 | 58.8 | 66.0 |
| Sketch | ViT-S | 49.7 | 40.9 | 48.1 | 26.1 | 47.8 | - | 42.5 | 49.7 |

Table 9: **Single-Source Transfer Performance on Office-Home.** The details accuracy information of the "Single-Source-∗" lines of Table 2.

| Target Domain | Backbone | Source Domain | | | | | |
|---|---|---|---|---|---|---|---|
| | | Art | Clipart | Product | Real World | Avg | Best |
| Art | ViT-S | - | 61.7 | 61.1 | 72.9 | 65.2 | 72.9 |
| Clipart | ViT-S | 49.8 | - | 49.4 | 60.9 | 53.3 | 60.9 |
| Product | ViT-S | 68.8 | 73.7 | - | 80.7 | 74.4 | 80.7 |
| Real World | ViT-S | 70.9 | 72.3 | 74.8 | - | 72.7 | 74.8 |

Table 10: **Single-Source Transfer Performance on Office-Home of LoRA.** The details accuracy information of the "Single-Source-∗" lines of Table 6. ViT-B backbone is already frozen and equipped with small trainable LoRA layers.

| Target Domain | Backbone | Source Domain | | | | | |
|---|---|---|---|---|---|---|---|
| | | Art | Clipart | Product | Real World | Avg | Best |
| Art | ViT-B | - | 68.4 | 74.4 | 73.8 | 72.2 | 74.4 |
| Clipart | ViT-B | 58.3 | - | 59.6 | 61.8 | 59.9 | 61.8 |
| Product | ViT-B | 82.0 | 80.8 | - | 84.9 | 82.6 | 84.9 |
| Real World | ViT-B | 81.0 | 80.3 | 81.9 | - | 81.0 | 81.9 |

### D.4 Baselines experiments settings in Table 2.

Since the significant difference from unsupervised methods and supervised methods, the results of MSFDA[21], DATE[9] and M3SDA[19] are directly supported by their own article.

On all the experiments of the baselines, we take all the source samples of different domains from trainset into account. And types of the backbone are all pretrained on ImageNet21k[5].

As for based on model-parameter-weighting few-shot methods: MCW[12] and H-ensemble[31], since they have not taken experiments on DomainNet and Office-Home datasets with ViT-Small backbone and 10-shot per class training samples, we take it by changing the backbone and samples condition while maintaining other configurations supported by their own work. And for fairness and efficiency, the well-trained source models from different domains of these methods are directly equipped with the same models trained by the first stage of our OTQMS method respectively.

As for based on samples few-shot methods: We exam the WADN[22] method under the condition of limited label on target domain as this setting is similar to ours. And we also change the backbone and samples condition while maintaining other configurations of its report to realize the experiments on DomainNet and Office-Home datasets. It is apparent that the settings of MADA[32] are quite different of ours, leveraging all the labeled source data and all the unlabeled target data with the few-shot labeled ones that are difficult to classify, which means all the target samples have been learned to some extent. So we not only decrease the labeled target samples to 10-shot per class but also the unlabeled target samples. And since MADA is the most SOTA and comparable method, we realize it with our fair 10-shot setting under both ViT-S and ResNet50 backbone on DomainNet and Office-Home datasets while maintaining other configurations of its report.

### D.5   Data efficient test details in 5.5 .

Table 11: **Data Efficient Test Results on DomainNet.** The capital letters represent the target domains.

| Method | Backbone | Data Counts ($\times 10^6$) | | | | | | | Training Consumed Time ($\times 10^4$ Second) | | | | | | |
|---|---|---|---|---|---|---|---|---|---|---|---|---|---|---|---|
| | | C | I | P | Q | R | S | Avg | C | I | P | Q | R | S | Avg |
| MADA[32] | ViT-S | 8.70 | 13.39 | 8.31 | 4.02 | 19.76 | 21.32 | 12.58 | 6.40 | 12.52 | 6.29 | 3.22 | 13.19 | 18.18 | 9.97 |
| MADA[32] | ResNet50 | 7.83 | 8.21 | 4.57 | 3.69 | 4.35 | 8.78 | 6.24 | 7.56 | 9.58 | 4.56 | 4.05 | 4.21 | 9.58 | 6.59 |
| AllSources $\cup$ Target | ViT-S | 2.17 | 1.42 | 1.42 | 4.20 | 1.42 | 1.90 | 2.09 | 0.56 | 0.37 | 0.39 | 1.09 | 0.36 | 0.49 | 0.54 |
| OTQMS (Ours) | ViT-S | 1.18 | 1.15 | 0.89 | 0.97 | 1.19 | 1.16 | 1.09 | 0.35 | 0.33 | 0.28 | 0.34 | 0.47 | 0.36 | 0.35 |

Table 12: **Data Efficient Log Scaled Test Results on DomainNet.** The capital letters represent the target domains.

| Method | Backbone | Data Counts (log scale) | | | | | | | Training Consumed Time (log scale) | | | | | | |
|---|---|---|---|---|---|---|---|---|---|---|---|---|---|---|---|
| | | C | I | P | Q | R | S | Avg | C | I | P | Q | R | S | Avg |
| MADA[32] | ViT-S | 15.98 | 16.41 | 15.93 | 15.21 | 16.80 | 16.87 | 16.35 | 11.07 | 11.74 | 11.05 | 10.38 | 11.79 | 12.11 | 11.51 |
| MADA[32] | ResNet50 | 15.87 | 15.92 | 15.33 | 15.12 | 15.29 | 15.99 | 15.65 | 11.23 | 11.47 | 10.73 | 10.61 | 10.65 | 11.47 | 11.10 |
| AllSources $\cup$ Target | ViT-S | 14.59 | 14.17 | 14.18 | 15.25 | 14.17 | 14.46 | 14.55 | 8.63 | 8.21 | 8.21 | 9.30 | 8.19 | 8.50 | 8.59 |
| OTQMS (Ours) | ViT-S | 13.98 | 13.95 | 13.70 | 13.78 | 13.99 | 13.96 | 13.90 | 8.17 | 8.10 | 7.93 | 8.12 | 8.46 | 8.18 | 8.17 |

### D.6   Domain choosing analysis details.

To compute the heatmap matrix visualizing domain preference in Figure 5(b), for each source domain, we count the samples selected from it until the target model converges under the 10-shot condition. Since the quantities of available samples varies significantly across domains, we normalize the counts by these quantities. The final domain preference is then determined by computing the importance of these normalized values.

Similarly, to compute the heatmap matrix for domain selection in Figure 5(a), we calculate the importance of the counts of selected samples from different source domains throughout the training epochs.

# E  Method to get $s^*$ and $\underline{\alpha}^*$ which minimize (14) in Theorem 7

The minimization problem of proposed measure (14) is

$$(s^*, \underline{\alpha}^*) \leftarrow \underset{(s, \underline{\alpha})}{\arg\min} \, \frac{d}{2} \left( \frac{1}{N_0 + s} + \frac{s^2}{(N_0 + s)^2} \frac{\underline{\alpha}^T \Theta^T J(\underline{\theta}_0) \Theta \underline{\alpha}}{d} \right). \tag{72}$$

We decompose this problem and explicitly formulate the constraints as follows.

$$(s^*, \underline{\alpha}^*) \leftarrow \underset{s \in [0, \sum\limits_{i=1}^{K} N_i]}{\arg\min} \, \frac{d}{2} \left( \frac{1}{N_0 + s} + \frac{s^2}{(N_0 + s)^2 d} \underset{\underline{\alpha} \in \mathcal{A}(s)}{\arg\min} \underline{\alpha}^T \Theta^T J(\underline{\theta}_0) \Theta \underline{\alpha} \right), \tag{73}$$

where

$$\mathcal{A}(s) = \left\{ \underline{\alpha} \, \middle| \, \sum_{i=1}^{K} \alpha_i = 1, s * \alpha_i \leq N_i, \alpha_i \geq 0, i = 1, \ldots, K \right\}. \tag{74}$$

Due to the complex constraints between $s$ and $\alpha$, obtaining an analytical solution to this problem is challenging. Therefore, we propose a numerical approach to get the optimal solution.

This problem requires optimizing the objective function over two variables: a scalar variable $s$ representing the total transfer quantity, and a vector variable $\underline{\alpha}$ representing the proportion of samples drawn from each source domain. For $s$ which is restricted to integer values, we perform a exhaustive search over its feasible domain $[0, s_{\max}]$, where $s_{\max} = \sum\limits_{i=1}^{K} N_i$. For each candidate $s'$ in the search, we compute the optimal $\underline{\alpha}'$ under the constraint $\mathcal{A}(s')$, which is a $K \times K$ quadratic programming problem with respect to $\underline{\alpha}$

$$\underline{\alpha}' = \arg \min_{\underline{\alpha} \in \mathcal{A}(s')} \underline{\alpha}^T \Theta^T J(\underline{\theta}_0) \Theta \underline{\alpha}.$$

The quadratic coefficient matrix in this optimization problem is given by $\Theta^\top J(\underline{\theta}_0) \Theta$. Since the Fisher information matrix $J(\underline{\theta}_0)$ is positive semi-definite, the quadratic coefficient matrix is also positive semi-definite. This guarantees the existence of a global optimal solution. After getting $s'$ and $\underline{\alpha}'$, the function is then evaluated at $(s', \underline{\alpha}')$.

In brief, for each $s'$, we solve for the corresponding optimal $\underline{\alpha}'$, yielding a finite collection of candidate solutions $(s', \underline{\alpha}')$ and their associated objective function values. After completing the search, the optimal solution $(s^*, \underline{\alpha}^*)$ is chosen as the pair that achieves the lowest objective function values among all candidate $(s', \underline{\alpha}')$. Since the feasible set of $s$ is finite and enumerable, and for each fixed $s'$ the optimization over $\underline{\alpha}'$ has a solution, the overall optimization problem is guaranteed to have at least one global solution. Hence, the optimal pair $(s^*, \underline{\alpha}^*)$ exists.

It is worth noting that, for the sake of computational efficiency, we do not exhaustively enumerate all possible values of $s$ in our experiments. Instead, we perform a grid search using 1000 uniformly spaced steps over the feasible range of $s$, where the number of steps is denoted as $stepnumber = 1000$. Experimental results in Section 5 demonstrate that this strategy does not compromise the effectiveness or stability of the method.

# F Analysis on Domain-specific Transfer Quantity

To understand the domain preference of OTQMS, we visualize the proportion of each source in the selected samples for each target domain. As shown in Figure 5(b), when the target domain is Clipart, OTQMS primarily leverages samples from Real, Painting, and Sketch. In addition, Quickdraw contributes minimally to any target domain. These observations align with the findings in [19]. To further clarify the selection process of OTQMS during training, we visualize it in Figure 5(a). We observe that in the `Office-Home` dataset, Clipart initially selects all source domains but later focuses on Art and Real World. In the `DomainNet` dataset, Sketch predominantly selects Clipart, Painting, and Real throughout the training process.

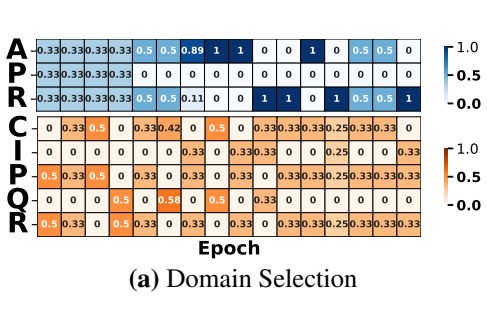

**(a)** Domain Selection

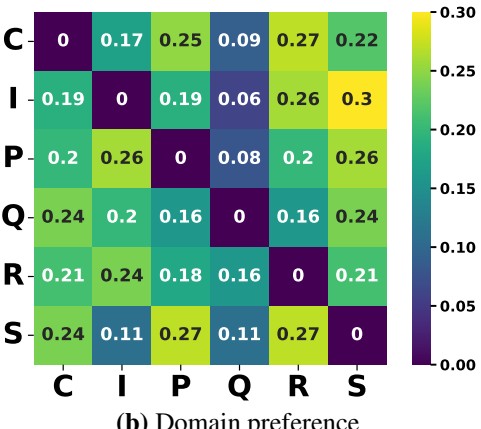

**(b)** Domain preference

Figure 5: Visualization of domain-specific transfer quantity under 10-shot setting. (a) Domain selection during training epochs (from left to right), where the blue upper part represents the selection of target domain Clipart on `Office-Home`, and the orange lower part represents the selection of target domain Sketch on `DomainNet`. Darker colors indicate stronger tendencies throughout the training process. (b) Source domain preferences of different target domains on `DomainNet`. Each row corresponds to a target domain while each column represents a source domain.

We additionally report the *raw average transfer quantities* between domains on the Office-Home and DomainNet datasets. Each row denotes a target domain and each column denotes a source domain.

Table 13: Average transfer quantities on the Office-Home dataset. Each row denotes a target domain and each column denotes a source domain.

| Target \ Source | A | C | P | R |
|---|---|---|---|---|
| A | 0 | 2546 | 2956 | 3341 |
| C | 1325 | 0 | 753 | 2192 |
| P | 736 | 1320 | 0 | 2378 |
| R | 1879 | 3371 | 3300 | 0 |

Table 14: Average transfer quantities on the DomainNet dataset. Each row denotes a target domain and each column denotes a source domain.

| Target \ Source | C | I | P | Q | R | S |
|---|---|---|---|---|---|---|
| C | 0 | 18391 | 57901 | 47641 | 138497 | 43100 |
| I | 28962 | 0 | 28979 | 17581 | 138497 | 41558 |
| P | 30907 | 41380 | 0 | 41945 | 83092 | 44356 |
| Q | 12380 | 11305 | 12883 | 0 | 30498 | 18050 |
| R | 24168 | 31066 | 36179 | 51749 | 0 | 27716 |
| S | 23591 | 11302 | 36897 | 50182 | 88133 | 0 |

# G   A Detailed Discussion on the Limitations

- **Sampling Method.** Since our theoretical analysis focuses on the transfer quantity from each source task, we adopt a straightforward random sampling strategy in the algorithm implementation. Given that our theoretical results are derived under average-case assumptions, random sampling is sufficient to demonstrate the robustness of both our theoretical analysis and the proposed algorithm. Nevertheless, we anticipate that more sophisticated sampling strategies, such as active sampling, may further improve the algorithm's performance.

- **Weight or Quantity.** Many existing work of multi-source transfer learning assign weights to source tasks and utilize all available samples. In contrast, this work focuses on optimizing the transfer quantity from each source task. We anticipate that future work could further improve algorithmic performance by jointly optimizing both the sample weights and the transfer quantity from each source task.

- **Possible Extensions to Other Loss Functions.** Our theoretical analysis is developed under the assumption of a negative log-likelihood objective, which aligns with the cross-entropy loss commonly used in classification tasks with softmax outputs and one-hot labels. However, for other learning objectives such as the mean squared error in regression problems, our framework does not yet provide a direct theoretical guarantee. We believe, nevertheless, that the core idea can be generalized to broader loss functions under suitable regularity conditions. We leave a more rigorous theoretical extension and empirical validation with alternative loss functions as future work to demonstrate the robustness and wider applicability of our method.

# H   Broader Impacts

We first develop a theoretical framework for optimizing transfer quantities in transfer learning, and subsequently propose an architecture-agnostic and data-efficient algorithm based on this theoretical framework. The proposed theoretical framework and algorithm have broad applicability in various transfer learning scenarios, including domains such as medical image analysis, recommendation systems, and anomaly detection. There are no negative social impacts foreseen.

