# OpenReview forum: "A High-Dimensional Statistical Method for Optimizing Transfer Quantities in Multi-Source Transfer Learning"
_NeurIPS.cc/2025/Conference — NeurIPS 2025 poster_

### Official Review · Reviewer_2Vcv · 2025-06-17

**Clarity:** 2
**Significance:** 2
**Originality:** 3
**Rating:** 4
**Confidence:** 4

**Summary:**

This paper aims to answer the question of “what is the optimal sample size for each sources when conducting transfer learning.” The authors proposes a parametric model-agnostic framework to find out the optimal size for each source by minimizing the expected KL divergence between true target distribution and the pooled-version estimated distribution w.r.t. source sample sizes. Practical algorithm is presented and the effectiveness of the proposed algorithm is evaluated through experiments.

**Questions:**

The authors provide a numerical approach in Appendix F to get the solution to objective (14). The authors proposed to do the grid search by first fixing $s$ in some feasible domain of $s$. However, what is the feasible domain of $s$ in general? What is it order? If no prior knowledge kicks in, the range of the feasible domain of $s$ can go quite large and ruin the computation.

**Ethical Concerns:**

["NO or VERY MINOR ethics concerns only"]

**Final Justification:**

After reading authors' response and other reviewers' comment, I will raise my score. Please make sure to revised the camera-ready version based on the discussion period accordingly.

**Limitations:**

Yes, although authors only put limitations in the appendix.

**Quality:**

2

**Strengths And Weaknesses:**

## Strength

1. To my knowledge, while most existing transfer learning work focus on finding the optimal sources, this paper goes with another direction by focusing on finding the optimal sample size the algorithm should use from each source.
2. The methodology is theory-driven.

## Weakness

1. **Terminology**: In theoretical statistics or machine learning community, the term “high-dimensional” usually refer to the cases where the dimension of the data fast excess the sample size, i.e., $d \gg n$, or the dimension grows as sample size increase. In such regime, many theoretical results in fixed dimension no longer hold and require different techniques or mathematical tools. For example, the asymptotic normality, i.e., eq (2), could fail under high-dimensional regime. I believe what authors consider is a fixed dimension $d$, which aligns with the classical multivariate analysis. Using “high-dimensional” is rather misleading.
2. **The set-up**: Although I don’t appreciate the idea that the authors consider the data are generated from a specific network parameterized by $\theta$ cuz it can produce some specification problem, it generally does not affect the core idea of the paper. However, the assumption that all source tasks and the target task share the same parametric model is worrying, which is practically impossible. This assumption is essential for the derivation in Section 4, as if it does not hold, the proposed methodology does not stand.
3. **The estimator**: The transfer learning estimator considered in this manuscript is rather simple, which is the minimizer of the pooled log-likelihood. It has been too simple to provide helpful insight into the current transfer learning paradigm, i.e., pre-training and fine-tuning. Pooling both source and target data to obtain an estimator is never effective and efficient, as has been long demonstrated in the community.

    Generally, I can understand why authors need to consider such estimator. This pooled version allows it corresponding distribution be the weighted combination of all source and target distribution (where the weights are determined by the sample size ratio). This allows the authors to derive the expression of expected KL divergence easier under the (1) same parametric model assumption (point 2) and (2) sufficient close models assumption (line 139). However, with so many restriction, again, it is too hard to have the theory provide insights.

4. **Assumption**:  The bounds in this paper are asymptotic and the authors require a “sufficient small models” assumption (line 139 or line 145) when derive these asymptotic results. However, in asymptotic regime, the “sufficient small models” actually indicates the source and target tasks/models are identical. Therefore, it is not surprised why the assumptions on same parametric model and pooled estimator are needed.
5. **Practical algorithm**: The practical algorithm (QTAMS) replaced the true source parameters by their empirical version. While this is a compromise with reality where the true parameters are unknown in practice, we have to remember that, since the bounds are asymptotic, there could be error terms induced by the approximation that are not lower order of $1/(N_0 + n_1)$ in eq (8), (12), (14). This will make the optimal solution for $n_{k}$ no longer follows the solutions derived from the objective in eq (8), (12), (14).
6. **Experiments**: I do not closely work on applications, so I typically have no idea what SOTA results would be, therefore, please view my comments on this point dialectically. For me, QTQMS only improved very limited compared to existing approaches/baselines, i.e., the improvement over the runner-up is rather small. What is more interesting to me is that the efficiency improved by QTQMS.

---

> ### Author Rebuttal · Authors · 2025-07-27
>
> We appreciate your detailed comments. Below, we address each of your concerns point by point.
>
> >Question1:what is the feasible domain of s?
>
> Thank you for the question. The feasible domain of the total transfer quantity $s$ is defined as the interval from 0 to the total number of available source samples, i.e., $s\in[0, s_{\max}]$ where $s_{\max} = \sum\limits_{i=1}^{K} N_i$. In our experiments, we perform a uniform grid search over this interval using 1000 equally spaced steps (denoted as $stepnumber = 1000$). The computational complexity of this procedure depends solely on the value of $stepnumber$, and is independent of the size of the feasible domain. Therefore, a large feasible range does not pose a challenge to the efficiency of the algorithm. A detailed analysis of the computational complexity of this process can be found in our response to Reviewer 43HZ, Question 4.
>
>
> >Weakness1: Using “high-dimensional” is rather misleading.
>
> Thank you for your suggestion. We agree that the term “high-dimensional” is misleading in our context, as our analysis is conducted in a fixed-dimensional regime. In the revised version of the paper, we will remove the term "high-dimensional statistics" and replace it with the more precise term "asymptotic analysis".
>
> >Weakness2: The assumption that all source tasks and the target task share the same parametric model is worrying, which is practically impossible.
>
> Thank you for raising this important concern. We would like to clarify that our framework only assumes that the source and target tasks share the same parametric model class, but each task is associated with its own set of model parameters. As described in Section 3, the underlying model for the target task is $P\_{X;\underline{\theta}\_0}$, while the source tasks $\mathcal{S}\_1, \dots, \mathcal{S}\_K$ correspond to models $P\_{X;\underline{\theta}\_1}, \dots, P\_{X;\underline{\theta}\_K}$, respectively.
>
> In transfer learning, it is common to select source and target tasks with the same input/output structure and similar feature spaces. This makes the use of a shared parameterized model a reasonable assumption, which is also widely adopted in the multi-source transfer learning literature [1][2].
>
> Moreover, we do not impose any specific structural assumptions on the form of the underlying model $P_{X;\underline{\theta}}$. In fact, by the Universal Approximation Theorem [5], a feedforward neural network with sufficient width can approximate any continuous function on a compact domain. This ensures that our framework is applicable to a broad class of tasks and models, without being limited to a particular architecture.
>
> In summary, while our theoretical derivation requires a shared model class, it allows full flexibility in parameterization across tasks. This assumption enables a general and tractable formulation, and our experiments further demonstrate the robustness and effectiveness of the proposed algorithm in realistic settings.
>
> >Weakness3:It has been too simple to provide helpful insight into the current transfer learning paradigm, i.e., pre-training and fine-tuning. Pooling both source and target data to obtain an estimator is never effective and efficient, as has been long demonstrated in the community.
>
> Thank you for the comment. We respectfully clarify that the model-based paradigm you mentioned (e.g., fine-tuning) and the sample-based paradigm adopted in this paper represent two established approaches to transfer learning. While model-based methods have been more extensively studied, recent works—as well as our own results—demonstrate that sample-based strategies can also achieve competitive performance [3][4], both effectively and efficiently when properly designed.
>
> We would also like to draw your attention to Table 2, where the runner-up method “AllSources $\cup$ Target” is a simple pooled estimator that uses all available source and target data for training. While naive, this baseline performs surprisingly well on certain datasets. The fact that more complex baselines do not consistently outperform such a simple pooling strategy suggests that they may not be universally suitable across all datasets.
>
> >Weakness4:The bounds in this paper are asymptotic and the authors require a “sufficient small models” assumption (line 139 or line 145) when derive these asymptotic results. However, in asymptotic regime, the “sufficient small models” actually indicates the source and target tasks/models are identical.
>
> Thank you for the comment. We would like to clarify that the assumption of a “sufficiently small model distance” (i.e., $|\theta_0 - \theta_1| = O(1/\sqrt{N_0})$) is a technical condition introduced to control higher-order terms in the asymptotic analysis. Given that the source and target tasks used in transfer learning often share similar input-output structures and semantic spaces, this assumption is also reasonable in practice.
>
> Nevertheless, it does not imply that the source and target tasks are identical. Importantly, even under this assumption, our framework allows for meaningful differences between tasks. If the tasks were truly identical, all source samples would always benefit the target task. In contrast, both our theoretical analysis (lines 155–164) and experimental results (Section 5) demonstrate that using all available source samples is not always optimal. Furthermore, in the multi-source setting, the optimal transfer quantity from each source depends explicitly on its distance to the target task, as formally characterized in Theorem 7 and empirically validated in Appendix G.
>
> >Weakness5:Practical algorithm: The practical algorithm (QTAMS) replaced the true source parameters by their empirical version. While this is a compromise with reality where the true parameters are unknown in practice, we have to remember that, since the bounds are asymptotic, there could be error terms induced by the approximation.
>
> Thank you for pointing out this important issue. We fully agree that the substitution of true source parameters $\theta_i$ with their empirical estimates $\hat{\theta}_i$ introduces approximation error. We acknowledge that this is a limitation of our current formulation, as the true source parameters are inaccessible in real-world scenarios. Despite this, we argue that our framework still provides a principled approximation scheme, and our empirical results (Section 5.2) show that it yields consistently strong performance across diverse settings. We will explicitly state this limitation in the revised version of paper. A more refined theoretical treatment that accounts for estimation error in source parameters is indeed an interesting direction for future work.
>
> Moreover, it is important to note that our definition of $N_i$ refers only to the number of source samples available during the target training stage, rather than the full dataset used to obtain the empirical estimates $\hat{\theta}_i$. In many real-world scenarios—such as the use of large pretrained models including GPT-3 [6] and PaLM [7]—the parameters $\hat{\theta}_i$ are obtained by training on datasets that are significantly larger than the accessible source sample sizes $N_i$, often incorporating non-public or proprietary data sources. In such cases, the estimation error of $\hat{\theta}_i$ is smaller than the order of $1 / (N_0 + n_1)$, making it a highly reliable approximation to $\theta_i$ and thereby justifying our substitution.
>
> >Weakness6:The improvement over the runner-up is rather small. What is more interesting to the reviewer is that the efficiency improved by QTQMS.
>
> Thank you for your thoughtful and balanced comment. We acknowledge that the performance gains of QTQMS over the strongest baselines are relatively modest in some cases. However, our goal is not only to pursue absolute accuracy, but also to develop a theoretically principled and practically efficient framework for multi-source transfer learning.
>
> In fact, several components of our current implementation remain intentionally simple, in order to better isolate and verify the robustness of the theoretical framework. For instance, after obtaining the optimal transfer quantity for each source domain, we adopt a straightforward random sampling strategy to construct the joint dataset. This design choice allows us to focus on validating the effectiveness of the core theory. Nevertheless, we believe that incorporating more advanced sampling strategies—such as active sampling—could further improve the performance of the algorithm.
>
> In addition, as shown in Section 5.4, QTQMS exhibits stable and superior performance across different shot settings, further demonstrating the robustness of the proposed method. As the reviewer kindly noted and as demonstrated in Section 5.5, QTQMS consistently achieves comparable or better accuracy while significantly reducing computational overhead, which we believe is one of the key advantages of our approach.
>
> Thank you again for your valuable insights. We kindly request that the reviewer considers these points when assigning the final scores.
>
> ## References：
> [1]Wu, Y., Wang, J., Wang, W., and Li, Y. H-ensemble: An information theoretic approach to reliable few-shot multisource-free transfer. AAAI 2024
>
> [2]Li Y, Yuan L, Chen Y, et al. Dynamic transfer for multi-source domain adaptation[C] CVPR2021
>
> [3]Shui C, Li Z, Li J, et al. Aggregating from multiple target-shifted sources[C]//International conference on machine learning. PMLR, 2021: 9638-9648.
>
> [4]Zhang W, Lv Z, Zhou H, et al. Revisiting the domain shift and sample uncertainty in multi-source active domain transfer[C] CVPR 2024
>
> [5] Hornik K. Approximation capabilities of multilayer feedforward networks[J]. Neural networks, 1991
>
> [6] Brown T, Mann B, Ryder N, et al. Language models are few-shot learners[J]. NeurIPS 2020
>
> [7] Chowdhery A, Narang S, Devlin J, et al. Palm: Scaling language modeling with pathways[J]. Journal of Machine Learning Research, 2023, 24(240): 1-113.

---

> > ### Comment · Reviewer_2Vcv · 2025-08-06
> >
> > First I would like to thanks authors for the detailed response and explanation.
> >
> > After reading it, my overall judgement will stay the same. A few reasons:
> >
> > **1**: The revision for this manuscript could be too much, including terminology changing and discussion on limitations of current formulation, theory and methodology.
> >
> > **2**: Some assumptions/settings are strong. For example, using real application setting to intuitively states the additional higher order terms is negligible is not enough for a theoretical driven paper. Besides, the term ``identical'' in my original comments does not mean they are the same (truly identical). It means in the asymptotic regime, the difference between source and target parameters can be very close 0, explaining why pooled MLE works and it is not surprise.
> >
> > **3**: Sample-level methods, like the pooled MLE estimator in this manuscript, can be very restrictive in practice in terms of storing/access both source and target data, require total retraining when switching to new tasks.

---

> ### Author Response · Authors · 2025-08-06
>
> We sincerely thank the reviewer for the time and effort spent in evaluating our work and for the constructive feedback provided throughout the discussion. We fully respect the reviewer’s final judgment regarding the score. The following clarification is offered not to dispute your assessment, but to further explain our reasoning and address potential misunderstandings.
>
> >Reason1: The revision for this manuscript could be too much
>
> We do not consider such revisions to be “too much”. In our discussion with you, only Weakness 1 and Weakness 5 involve modifications. Across the discussions with the other three reviewers, there are only two additional questions where we proposed modifications for the revised version. Moreover, the corresponding revised text has already been completed and presented in this rebuttal, and the only remaining work for the revised paper is the substitution of text. We also assure you and all other reviewers that all the modifications mentioned in rebuttal will be fully implemented in the final version of the paper.
>
> >Reason2: Some assumptions/settings are strong. Some theory are intuitive result using real application setting
>
> First, we would like to clarify that our reference to realistic scenarios serves to define our assumptions and research scope, ensuring that the framework remains applicable to real-world settings. However, our theoretical framework is derived from formal statistical principles under these assumptions, rather than from intuitive reasoning.
>
> For example, in our response to Weakness 5, we referred to real-world cases—such as the use of large pretrained models including GPT-3 and PaLM—where the parameters $\hat{\theta}_i$ are obtained by training on datasets whose sizes are significantly larger than the accessible source sample sizes $N_i$. This reference was intended solely to motivate the assumptions, but the conclusion that “the estimation error of $\hat{\theta}_i$ is smaller than the order of $1 / (N_0 + n_1)$” is firmly grounded in established statistical theory. Specifically, when the source parameters $\hat{\theta}_i$ are obtained from pretraining on $M_i \gg (N_0 + n_1)$ samples, the asymptotic MSE of the MLE scales as $O(1/M_i)$ using Lemma 1, which is negligible compared to the $O(1/(N_0 + n_1))$ term in our asymptotic bounds. Therefore, substituting $\hat{\theta}_i$ for $\theta_i$ does not introduce error terms of the same order, and this conclusion is derived from rigorous theoretical analysis rather than from simply asserting, based on real-world examples, that additional higher-order terms are negligible.
>
> For the “identical” assumption and the use of pooled MLE, we would like to clarify that our “identical” assumption—interpreted as a small source–target parameter distance—is not “strong”; rather, It is motivated by realistic transfer learning scenarios in which the source and target tasks typically share similar input–output structures and semantic spaces. The fact that the theory based on pooled MLE works in our experiments is because this assumption matches the characteristics of widely used real-world datasets such as Office-Home and DomainNet, making our theory applicable, rather than this “strong” assumption directly causing favorable experimental results. Importantly, having the parameter distance be asymptotically small does not imply the trivial conclusion that the pooled MLE using all available source samples is always optimal. On the contrary, both our theoretical analysis (lines 155–164) and experimental results (Section 5) demonstrate that using all available source samples is not always optimal.
>
> >Reason3: Sample-level methods can be very restrictive in practice
>
> First, we would like to reiterate that sample-based transfer learning constitutes a meaningful and active research direction, whose significance and effectiveness has been confirmed by several recent works [1][2][3]. Our own experimental results further demonstrate that this approach can perform competitively across diverse settings. We acknowledge that our method requires accessing source data; however, this limitation is inherent to a broad class of sample-based transfer learning approaches, and scenarios in which source data can be accessed do occur in practice. Second, our framework can be integrated with strategies such as LoRA (Section 5.6), which can greatly reduce retraining overhead by avoiding full-model updates when switching tasks.
>
> ## Reference:
>
> [1]Shui C, Li Z, Li J, et al. Aggregating from multiple target-shifted sources[C]//International conference on machine learning. PMLR, 2021: 9638-9648.
>
> [2]Zhang W, Lv Z, Zhou H, et al. Revisiting the domain shift and sample uncertainty in multi-source active domain transfer[C] CVPR 2024
>
> [3]Li D, Zhang Z, Wang L, et al. Scalable Fine-tuning from Multiple Data Sources: A First-Order Approximation Approach[C]// EMNLP 2024 Findings

---

### Official Review · Reviewer_ZUcy · 2025-06-20

**Clarity:** 3
**Significance:** 3
**Originality:** 3
**Rating:** 5
**Confidence:** 4

**Summary:**

This paper discusses the optimal quantity of source samples from each source task. A generalization error measure based on KL divergence is introduced. The quantity of source samples can be solved based on high-dimensional statistical analysis. Experiment results demonstrate the effectiveness of the proposed method.

**Questions:**

The questions are consistent with the "Weaknesses".

1.	As the learning objective of many problems are not limited to negative log likelihood, I am interested in the optimal quantities with general learning objectives, such as cross-entropy loss in classification problems and mean square error in regression problems, or at least some error analysis when the proposed method is applied in general settings.

2.	How can the proposed method combine with other parts of the transfer learning? For examples, the sample selection and domain alignment. I understand these may be a bit  out of the scope of this paper, so it's only a suggestion and the answer will not decrease the score.

**Ethical Concerns:**

["NO or VERY MINOR ethics concerns only"]

**Final Justification:**

Despite existing limitations, I still believe that the proposed method to optimize transfer quantity is a meaningful and novel contribution to the field of transfer learning.

**Limitations:**

The authors have mentioned the sampling method in the discussion of limitations, but the limitations in the general training objective have not been discussed.

**Quality:**

3

**Strengths And Weaknesses:**

Strengths:

1. The task that estimates the optimal quantity of source samples holds broad application scenarios. And the theoretical results are intuitive yet not trivial.

2. The derivation of the method is reasonable, and the training process is clear and is easy to generalize to other tasks.

3. The authors have conducted sufficient evaluation and analysis experiments, demonstrating the effectiveness and efficiency of the proposed method.

Weaknesses:

1. The main theorems are derived based on the MLE estimation of the model parameters. In learning problems with other objectives, the effectiveness of the results is unclear.

2. Optimizing the quantities of source samples should be considered as part of the whole transfer learning paradigm. Therefore, further discussions about the relation with sample selection and domain alignment will be helpful.

---

> ### Author Rebuttal · Authors · 2025-07-27
>
> We are grateful to you for taking the time to review our paper and provide thoughtful insights. Below, we address each of your concerns point by point.
>
>
> >Question1：As the learning objective of many problems are not limited to negative log likelihood, I am interested in the optimal quantities with general learning objectives, such as cross-entropy loss in classification problems and mean square error in regression problems, or at least some error analysis when the proposed method is applied in general settings.
>
> Thank you for your valuable suggestion. For the case of cross-entropy loss in classification problems, it is essentially equivalent to the negative log-likelihood when softmax outputs and one-hot encoded labels are used. This corresponds precisely to the setup adopted in the main experiments of our paper.
>
> For other learning objectives, such as mean squared error in regression tasks, we acknowledge that there is a gap compared to the negative log-likelihood formulation assumed in our theoretical analysis. While our framework does not directly cover these cases, we believe the core idea can be extended to broader loss functions under certain regularity conditions. We will clarify this limitation in the revised version of paper, and include additional experiments with other loss functions to empirically demonstrate the robustness and broader applicability of our method.
>
> >Question2：How can the proposed method combine with other parts of the transfer learning? For examples, the sample selection and domain alignment.
>
> Thank you for the insightful suggestion, which points to a valuable direction for future work. In this paper, after obtaining the optimal transfer quantity of each source domain, we adopt a straightforward random sampling strategy to construct the joint dataset in our algorithm implementation. As our theoretical analysis is based on average-case assumptions, random sampling is sufficient to validate the robustness of both the theoretical framework and the proposed algorithm. Nonetheless, we anticipate that more advanced sample selection strategies, such as active sampling, could further enhance the algorithm’s performance. We plan to explore this possibility in future work.
>
> Once again, we thank you for the constructive feedback.

---

> > ### Comment · Reviewer_ZUcy · 2025-08-05
> >
> > Thank you for the authors' response. After reading the rebuttal and the comments from other reviewers, I agree with the concerns raised regarding the theoretical assumptions. However, I still believe that the formulation of a method to optimize transfer quantity represents a meaningful and novel contribution to the field of transfer learning. Therefore, I will maintain my initial rating.

---

> > > ### Author Response · Authors · 2025-08-09
> > >
> > > We are truly grateful for your careful reading of our paper and the clarity of your feedback. Your professional insights have significantly contributed to enhancing the quality and clarity of our manuscript.

---

### Official Review · Reviewer_43HZ · 2025-06-25

**Clarity:** 4
**Significance:** 3
**Originality:** 3
**Rating:** 5
**Confidence:** 4

**Summary:**

This paper presents a novel theoretical framework for determining the optimal number of samples to transfer from each source task in a multi-source transfer learning setting. The authors formulate the problem as a parameter estimation task and leverage high-dimensional statistical analysis to derive the optimal transfer quantities by minimising a generalisation error measure based on Kullback-Leibler (K-L) divergence. This framework is implemented in a practical, architecture-agnostic algorithm named OTQMS. The method is empirically evaluated on standard benchmarks, demonstrating superior performance in both accuracy and data efficiency compared to existing approaches.

**Questions:**

My questions are related to the weaknesses mentioned in the critical review above. I will summarise my main concerns in the form of questions for the authors:
- The theoretical framework relies on the assumption that task parameters are asymptotically close ($|\theta_{0}-\theta_{1}|=O(\frac{1}{\sqrt{N_0}})$). While the empirical results appear robust, what are the theoretical implications when this assumption is strongly violated, for instance, in cases of far-domain transfer? How does the framework behave, and could it be extended to formally account for larger parameter distances?
- The framework models task divergence as a unified parametric distance ($\underline{\theta}_i - \underline{\theta}_0$), which aggregates different sources of domain shift. Can the framework be decomposed to provide more granular insights into how different types of shift (e.g., covariate shift vs. concept shift) individually impact the optimal transfer quantity?
- The dynamic OTQMS algorithm initializes the target parameter $\underline{\theta}_0$ using only the limited target data before iteratively refining it. How sensitive is the final model performance to the quality of this initial, and potentially noisy, estimation? Have you investigated whether a poor initialization could lead the optimization to a suboptimal solution?
- The paper highlights significant reductions in training time. However, the OTQMS algorithm introduces a selection step in each epoch that involves solving a constrained quadratic program. Could you provide an analysis of the computational overhead of this selection process itself, particularly as the number of source domains ($K$) increases?
- In the high-dimensional case, the task discrepancy term $t$ is an average over the parameter dimension $d$. Does this averaging potentially mask the impact of large discrepancies in a small subset of critical parameters? Would a dimension-weighted approach offer further improvements?
- On page 4, you state the assumption $||\theta_i-\theta_0||=O(1/\sqrt{N_0})$ is "supported by related studies [20]". Could you elaborate on how the cited work [20] specifically justifies this scaling law for parameter distance in the context of transfer learning?

**Ethical Concerns:**

["NO or VERY MINOR ethics concerns only"]

**Final Justification:**

I maintain my recommendation to accept this paper.

​The work addresses a novel and practical question in multi-source transfer learning: how much data to transfer from each source. The proposed theoretical framework, based on minimizing KL divergence, provides a principled motivation for the OTQMS algorithm. The algorithm itself is architecture-agnostic and demonstrates compelling empirical performance, outperforming several state-of-the-art methods on standard benchmarks.

​I have read the authors' rebuttal and find their responses to my questions satisfactory. They have adequately clarified the limitations of the theoretical assumption regarding parameter distance, provided a reasonable analysis of the computational overhead of their method, and addressed concerns about the algorithm's sensitivity to initialization. ​While the theoretical guarantees are confined to a specific regime, the paper's primary contribution lies in formulating this problem and providing an effective practical solution.

**Limitations:**

My main comments on the limitations of this work are detailed in the "Weaknesses" section of the review above. To summarise, the primary limitations are: (1) the restrictive theoretical assumption that source and target task parameters are asymptotically close, which may not hold in cases of significant domain shift; (2) the framework's inability to distinguish between different types of transfer (e.g., covariate vs. concept shift), limiting its explanatory power; and (3) unanalyzed sensitivities in the practical OTQMS algorithm, namely its dependence on a potentially noisy initial estimate of the target model and the unexamined computational overhead of the sample selection step.

**Quality:**

3

**Strengths And Weaknesses:**

# Strengths

- The paper addresses a critical and often overlooked question in transfer learning: not just which sources to use, but how much data to use from each.
- At its core, the paper proposes a solid theoretical analysis. It introduces the expected K-L divergence as a measure of generalisation error, which is well-justified by its connection to cross-entropy loss. The use of high-dimensional statistical analysis to derive the optimal transfer quantities is rigorous, and the proofs are clear.
- The theoretical findings are successfully translated into a practical algorithm, OTQMS. A key feature is the dynamic strategy that iteratively updates the estimate of the target parameter $\theta_0$​ and re-computes the optimal quantities, mitigating the issue of having limited target data. The algorithm is architecture-agnostic nature and its demonstrated compatibility with both full model training and parameter-efficient methods like LoRA.
- The authors conduct extensive experiments on multiple real-world datasets (DomainNet, Office-Home, Digits) and across various settings. OTQMS consistently outperforms a range of state-of-the-art baselines in accuracy.

# Weaknesses

- The theoretical derivation for the optimal transfer quantity in Theorem 4 and its extensions relies on the assumption that the parameter distance between source and target tasks is small, i.e., $||\theta_i-\theta_0||=O(1/\sqrt{N_0})$. The paper argues this is a reasonable assumption supported by related work and that its conclusions hold even when the distance is large. However, this is a strong condition that may not hold in cases of significant domain shift. The theoretical guarantees are strongest for closely related tasks, and the robustness in practice does not fully alleviate this theoretical limitation.
- The framework models the difference between tasks solely as a parametric distance, $\theta_i-\theta_0$, within a given model class $P(X|\theta)$​​. While this general formulation is a strength, it also prevents a deeper analysis of why tasks differ. The framework does not distinguish between different transfer learning scenarios, such as covariate shift (change in the input distribution $P(Z)$) versus concept shift (change in the conditional distribution $P(Y|Z)$). Consequently, it cannot provide insights into how these distinct types of domain shifts affect the optimal transfer quantity. The parameter vector $\theta$​ conflates these potentially disparate effects.
- The OTQMS algorithm requires computing the optimal transfer quantities $(s,\alpha)$ in each training epoch. This is achieved by solving a quadratic programming problem for $\alpha$ within a numerical search for $s$. While the paper reports a net reduction in training time due to using fewer samples, it does not analyse the computational overhead of this selection step itself. In scenarios with a very large number of source tasks (large $K$), this step could introduce a non-trivial computational burden that might offset the gains from training on a smaller dataset.
- The dynamic strategy for estimating the target parameter $\theta_0$ begins with a model trained only on the (often scarce) target samples. The accuracy of this initial estimate can be poor, potentially influencing the subsequent iterative optimisation process. While the dynamic approach is shown to be superior to a static one, the sensitivity of the final performance to this initial, potentially noisy, estimate of $\theta_0$ is not explored.

# Conclusion

The formulation of a method to optimise transfer quantity is a significant contribution, and the empirical results are compelling. However, the theoretical claims are narrower than the empirical results might suggest due to restrictive assumptions. I believe that the paper deserves acceptance for its novelty and practical impact, but its limitations should be clearly acknowledged. The theoretical framework is more of a good motivation for an optimisation strategy than a general theory of transferability.

---

> ### Author Rebuttal · Authors · 2025-07-27
>
> We sincerely appreciate the reviewer’s thoughtful observation. Below, we address each of your concerns point by point.
>
>
> >Question1：How does the framework behave, and could it be extended to formally account for larger parameter distances?
>
> Thank you for the valuable feedback. We agree that the assumption of a small parameter distance (i.e., $|\theta_0 - \theta_1| = O(\frac{1}{\sqrt{N_0}})$) is a strong condition underlying the asymptotic derivation in Theorem 4 and its extensions. This assumption is necessary for controlling higher-order terms in the theoretical analysis. Nevertheless, the optimal transfer quantity derived from the framework remains consistent with intuition even when the assumption does not strictly hold. In particular, as shown in lines 160–162, when the parameter distance becomes large, the optimal transfer quantity naturally reduces to zero, reflecting the expected behavior under significant domain shift. This suggests that the theoretical formulation still captures the correct transfer tendency even for less related tasks. Moreover, our empirical results in Section 5 demonstrate that the proposed algorithm remains robust across a wide range of transfer scenarios.
>
> >Question2：Can the framework be decomposed to provide more granular insights into how different types of shift (e.g., covariate shift vs. concept shift) individually impact the optimal transfer quantity?
>
> Thank you for the insightful comment. Our framework models task divergence as a unified parametric distance between $\underline{\theta}_0$ and $\underline{\theta}_i$, which implicitly captures different types of domain shift (e.g., covariate shift and concept shift). This design reflects our goal of constructing a task-agnostic framework that determines the optimal transfer quantity without relying on prior knowledge of task-specific structure or decomposition.
>
> On the other hand, in scenarios where such prior knowledge is available and the model architecture explicitly separates covariate and concept shifts, for example by assigning them to different layers for processing, the parametric difference vector $\underline{\theta}_0 - \underline{\theta}_i$ naturally reflects these distinctions  across different subsets of parameters. In such cases, our method can effectively incorporate both types of shift during optimization without requiring explicit separation.
>
> >Question3：How sensitive is the final model performance to the quality of this initial, and potentially noisy, estimation?
>
> As discussed in Section 5.4, we investigate the performance of OTQMS under different shot settings. The results show that increasing the number of target samples indeed improves the quality of the initial estimation, which in turn leads to better model performance. Nevertheless, across the vast majority of settings, our method consistently outperforms the baselines, demonstrating its robustness to the quality of the initial estimation.
>
> >Question4：The paper highlights significant reductions in training time. However, the OTQMS algorithm introduces a selection step in each epoch that involves solving a constrained quadratic program. Could you provide an analysis of the computational overhead of this selection process itself, particularly as the number of source domains $K$ increases?
>
> Thank you for the question. The detailed procedure of the selection step is provided in Appendix F. Here, we provide an analysis of its computational complexity. Specifically, the procedure involves optimizing over two variables: a scalar variable $s$ representing the total transfer quantity, and a vector variable $\underline{\alpha}$ representing the proportion of samples drawn from each source domain. We perform a uniform grid search over the feasible range of $s$, i.e., $[0, s_{\max}]$, using 1000 uniform steps (denoted as $stepnumber = 1000$), where $s_{\max}= \sum\limits_{i=1}^{K} N_i$. For each candidate value $s'$, we solve a constrained optimization problem over $\underline{\alpha}$ under the constraint set $A(s')$, which corresponds to a $K \times K$ quadratic program (QP) with complexity $\mathcal{O}(K^3)$. Thus, the overall time complexity of the selection process is $\mathcal{O}(stepnumber \cdot K^3)$. In our experiments on the Office-Home dataset, where $stepnumber = 1000$ and $K = 3$, the total computational cost of the selection step is approximately $2.7 \times 10^4$ basic operations per epoch. Empirically, the selection step takes around 3.19 seconds per epoch on average, accounting for only 1.1% of the total training time. We acknowledge the reviewer’s concern regarding the growth of computational overhead with respect to $K$. In practice, the number of source domains in standard transfer learning benchmarks is typically small (e.g., $K \leq 10$), making the $\mathcal{O}(stepnumber \cdot K^3)$ complexity acceptable for most practical applications. Moreover, for scenarios with a large number of source domains, one possible extension is to cluster or merge similar domains before applying our algorithm.
>
>
>
> >Question5：Does this averaging potentially mask the impact of large discrepancies in a small subset of critical parameters? Would a dimension-weighted approach offer further improvements?
>
> Under the high-dimensional setting in Eq. (13) and Eq. (15), the Fisher information matrix $J$ within the $t$ term naturally plays a role similar to dimension-weighted averaging. Specifically, the Fisher matrix captures the model’s sensitivity to perturbations in different dimensions of the parameter vector, thereby implicitly assigning greater importance to more influential parameters.
>
> >Question6：Could you elaborate on how the cited work [20] specifically justifies this scaling law for parameter distance in the context of transfer learning?
>
> Thank you for the question. [20] provides empirical evidence that supports this assumption. The paper shows that during fine-tuning, model parameters—especially in the lower layers—change very little from their pretrained values. This suggests that for related domains or similar data distributions, the corresponding optimal parameters lie close in parameter space. Since transfer learning typically involves source and target tasks with some degree of similarity, we interpret this as indirect support for our assumption that the parameter distance between source and target models is small in transfer learning settings.
>
> Thank you for your insightful comment!

---

> > ### Comment · Reviewer_43HZ · 2025-08-04
> >
> > After reading the authors' rebuttal and the comments of other reviewers, I don't have any follow-up questions and will maintain my initial rating.

---

> > > ### Author Response · Authors · 2025-08-09
> > >
> > > We sincerely appreciate the time and effort you invested in carefully reviewing our manuscript. Your insightful observations and professional perspective have been invaluable in helping us refine and strengthen our work.

---

### Official Review · Reviewer_rZbt · 2025-07-05

**Clarity:** 3
**Significance:** 4
**Originality:** 4
**Rating:** 4
**Confidence:** 3

**Summary:**

This paper tackles a fundamental problem in multi-source transfer learning (MSTL): how to eliminate negative transfer from a sample-level perspective. The authors aim to determine the optimal number of samples to transfer from each source task to best benefit the target task. Unlike existing approaches that transfer all available data or rely on heuristic selection strategies, the paper proposes a theoretically grounded framework for transfer quantity optimization.

Specifically, the authors introduce a generalization error measure based on the expected Kullback-Leibler (KL) divergence between the true target distribution and the learned model. They leverage asymptotic high-dimensional statistical analysis to derive closed-form expressions for the optimal transfer quantities in both single-source and multi-source settings. This theoretical foundation leads to the development of OTQMS, a data-efficient and architecture-agnostic algorithm. Furthermore, the paper proposes a dynamic training strategy that iteratively updates both the estimation of the target model parameters and the corresponding optimal transfer quantities during training.

**Questions:**

1. Fisher Information Estimation:
Given that the Fisher information matrix is estimated using only a small number of target samples, how do the authors ensure the stability and reliability of this estimate? Have they evaluated the sensitivity of the algorithm to noise or variance in the empirical Fisher?

2. Negative Transfer Evidence:
While the theoretical analysis suggests that transferring too many source samples may lead to negative transfer, the experimental results do not explicitly show this. Can the authors provide empirical evidence demonstrating performance degradation when the transfer quantity exceeds the theoretical optimum?

3. Optimal Transfer Quantity in Practice:
The framework is designed to compute the optimal number of samples to transfer from each source, but the paper does not state what these optimal quantities actually are in practical settings. Could the authors report or visualize the learned transfer quantities to enhance interpretability?

4. Existence and Solvability of the Optimum:
Has the existence and uniqueness of the solution to the optimization problem in Theorem 7 been formally analyzed or empirically verified? Are there conditions under which the solution may not exist?

**Ethical Concerns:**

["NO or VERY MINOR ethics concerns only"]

**Final Justification:**

I appreciate the authors' efforts in addressing my questions. For Q1, the stability of the Fisher information estimate in the initial stage appears to be demonstrated only through empirical results, but I think its robustness remains unclear. Nevertheless, the proposed method demonstrates clear novelty and technical soundness, and the authors have sufficiently addressed our other concerns (Q2–Q4).

Given the overall quality of the work and the thoughtful responses, I have decided to raise our score.

**Limitations:**

One key limitation of the proposed framework is its reliance on asymptotic statistical assumptions, which may not hold in extreme few-shot regimes. In such cases, the empirical estimation of the Fisher information matrix based on a small number of target samples can suffer from high variance and numerical instability, potentially affecting the reliability of the computed transfer quantities

**Quality:**

3

**Strengths And Weaknesses:**

Strength

The paper presents a theoretically grounded formulation for optimizing transfer quantities in multi-source transfer learning. By leveraging KL-divergence-based generalization error and asymptotic high-dimensional statistical analysis, it offers a novel and mathematically rigorous approach. A key strength of the work lies in its focus on sample-level optimization, which allows for fine-grained control over transfer and directly addresses the problem of negative transfer, unlike prior approaches that operate at the domain or model level. The proposed algorithm, OTQMS, is both architecture-agnostic and data-efficient, enabling effective knowledge transfer while minimizing unnecessary sample usage.
Empirically, OTQMS demonstrates strong performance on challenging benchmarks such as DomainNet and Office-Home with notable improvements in accuracy. The method exhibits robustness across a wide range of shot settings from 5-shot to 100-shot.

Weakness

While the proposed framework is theoretically well-grounded, it relies on asymptotic analysis, and the paper does not offer non-asymptotic guarantees or finite-sample bounds. This limits our understanding of how well the method performs when the number of target samples is extremely limited. A central step in the algorithm is estimating the Fisher information matrix using a small set of target samples. While this is a practical design choice, it raises concerns about the variance and numerical stability of the estimate—especially in few-shot settings where data scarcity may result in unreliable curvature information. Furthermore, although the authors argue that transferring too many samples can lead to negative transfer, the experiments do not clearly show this behavior. The paper does not quantify what the optimal number of transferred samples actually is, nor does it provide empirical evidence of performance degradation when that number is exceeded. This weakens the practical insight into when and how negative transfer occurs. In addition, the optimization problem in Theorem 7, involving the total transfer quantity \$s^*$  and the proportion vector \$\alpha^∗$, is said to be solved numerically, but Appendix F provides insufficient detail about how this is actually done. It is unclear whether a unique or stable solution always exists, and the lack of discussion on the solvability or initialization sensitivity may hinder reproducibility and theoretical completeness.

One minor notation issue: the paper uses underlines to denote high-dimensional vectors. It is more standard and clearer to use boldface notation to indicate vector quantities.

---

> ### Author Rebuttal · Authors · 2025-07-27
>
> Thank you for your insightful feedback on our paper. Below, we address each of your concerns point by point.
>
> >Q1:Fisher Information Estimation: Given that the Fisher information matrix is estimated using only a small number of target samples, how do the authors ensure the stability and reliability of this estimate?
>
>
> The dynamic strategy proposed in lines 210–212 can address this issue. Specifically, in the first epoch, we train a $\underline{\theta}_0$ using only the target data, and get the $J(\underline{\theta}_0)$ using the gradient of loss.
> This $\underline{\theta}_0$ and $J(\underline{\theta}_0)$ is then used to determine the optimal transfer quantity from each source task, and we use random sampling to form a new resampled training dataset. Finally, we continue training $\underline{\theta}_0$ and get new $J(\underline{\theta}_0)$ on this new dataset, and this procedure is repeated in each subsequent epoch to iteratively update the training dataset. In brief, only in the algorithm’s initial epoch do we use a small number of target samples to train the Fisher information matrix; in subsequent epochs, we incorporate additional source samples to assist in training the Fisher information matrix, and the inclusion of more samples helps ensure its stability. Introducing source samples to assist in training the Fisher information matrix can ensure its stability without incurring significant error, because, as stated in line~533, the difference between ${J}(\theta_0)$ and ${J}(\theta_1)$ is of the order $O\left(\frac{1}{\sqrt{N_0}}\right)$. We will include the proof of this result in the revision version of the paper.
>
> >Q2:Negative Transfer Evidence: While the theoretical analysis suggests that transferring too many source samples may lead to negative transfer, the experimental results do not explicitly show this. Can the authors provide empirical evidence demonstrating performance degradation when the transfer quantity exceeds the theoretical optimum?
>
> We have addressed this issue at two points in the paper. First, in Figure 1, we observe that when the quantity of target samples is large, training with both target task samples and all source samples may perform worse than using target task samples only. Second, in Table 2, the method "AllSources $\cup$ Target", which uses all available source and target data for training, gain worse performance than our proposed optimal transfer quantity method, OTQMS.
>
> Furthermore, to better address your concern, we conducted additional experiments. Specifically, over the feasible domain of the total transfer quantity $s \in [0, s_{\max}]$, where $s_{\max}= \sum\limits_{i=1}^{K} N_i$, we evaluated the training performance corresponding to the transfer quantity at each decile point of $s_{\max}$.
>  The results demonstrate that none of these transfer quantities achieve better performance than our proposed optimal transfer quantity $s^{*}$. This experiment is conducted on the Office-Home dataset, and the table reports the average accuracy across all domains when each is used as the target domain.
>
>
> | $s = 0$ | $s = 0.1s_{\max}$ | $s = 0.2s_{\max}$ | $s = 0.3s_{\max}$ | $s = 0.4s_{\max}$ | $s = 0.5s_{\max}$ | $s = 0.6s_{\max}$ | $s = 0.7s_{\max}$ | $s = 0.8s_{\max}$ | $s = 0.9s_{\max}$ | $s = s_{\max}$ | $s^{*} \approx 0.67s_{\max}$ |
> |:-------:|:-----------------:|:-----------------:|:-----------------:|:-----------------:|:-----------------:|:-----------------:|:-----------------:|:-----------------:|:-----------------:|:--------------:|:---------------------------:|
> | 45.2%   | 57.2%              | 67.1%              | 68.6%              | 70.1%              | 73.1%              | 75.2%              | 77.5%              | 77.3%              | 77.1%              | 77.2%          | 78.2%                        |
>
>
>
> >Q3:Optimal Transfer Quantity in Practice: The framework is designed to compute the optimal number of samples to transfer from each source, but the paper does not state what these optimal quantities actually are in practical settings. Could the authors report or visualize the learned transfer quantities to enhance interpretability?
>
> Your suggestion is very helpful. In Figure 5(b) of Appendix G, we provide a domain preference heatmap for transfer on the DomainNet dataset, which reflects the normalized transfer proportion of each source domain. Your comment made us realize the importance of also reporting the raw transfer quantities. We hereby supplement the actual average transfer quantities on both the Office-Home and DomainNet datasets, and will include them in the revised version of the paper. In the table below, each row corresponds to a target domain, while each column represents a source domain.
>
> | Office-Home | A    | C    | P    | R    |
> |-----|------|------|------|------|
> | A   | 0    | 2546 | 2956 | 3341 |
> | C   | 1325 | 0    | 753  | 2192 |
> | P   | 736  | 1320 | 0    | 2378 |
> | R   | 1879 | 3371 | 3300 | 0    |
>
>
>
>
> |  DomainNet   | C     | I     | P     | Q     | R      | S     |
> |-----|-------|-------|-------|-------|--------|-------|
> | C   | 0     | 18391 | 57901 | 47641 | 138497 | 43100 |
> | I   | 28962 | 0     | 28979 | 17581 | 138497 | 41558 |
> | P   | 30907 | 41380 | 0     | 41945 | 83092  | 44356 |
> | Q   | 12380 | 11305 | 12883 | 0     | 30498  | 18050 |
> | R   | 24168 | 31066 | 36179 | 51749 | 0      | 27716 |
> | S   | 23591 | 11302 | 36897 | 50182 | 88133  | 0     |
>
>
> >Q4:Existence and Solvability of the Optimum: Has the existence and uniqueness of the solution to the optimization problem in Theorem 7 been formally analyzed or empirically verified? Are there conditions under which the solution may not exist?
>
>
> Here, we provide a more detailed explanation of the optimization method of Theorem 7 described in Appendix F, and prove the existence of a globally optimal solution. The minimization problem of the objective function in Theorem 7 is Eq. (74), i.e.,
>
> $$
> (s^\*, \underline{\alpha}^\*) \gets \arg \min_{(s, \underline{\alpha})} \frac{d}{2}\left(\frac{1}{N_0+s}+\frac{s^2}{(N_0+s)^2}\frac{\underline{\alpha}^T\Theta^T{J}({{\underline{\theta}}_0})\Theta\underline{\alpha}}{d}\right).
> $$
>
> We decompose this problem and explicitly formulate the constraints as Eq. (75), i.e.,
>
> $$
> (s^\*, \underline{\alpha}^\*) \gets \arg\min_{s\in[0,\sum\limits_{i=1}^{K}N_i]} \frac{d}{2}\left(\frac{1}{N_0+s}+\frac{s^2}{(N_0+s)^{2}d}\arg\min\limits_{\underline{\alpha}\in\mathcal{A}(s)}\underline{\alpha}^T\Theta^T{J}({{\underline{\theta}}_0})\Theta\underline{\alpha}\right),
> $$
>
> where
>
> $$
> \mathcal{A}(s)=\bigg\\{\underline{\alpha}\Bigg|\sum_{i=1}^{K}\alpha_i=1, s*\alpha_i \le N_i, \alpha_i \ge0,i=1,\dots,K \bigg\\}.
> $$
>
> This problem requires optimizing the objective function over two variables: a scalar variable $s$ representing the total transfer quantity, and a vector variable $\underline{\alpha}$ representing the proportion of samples drawn from each source domain. For $s$ which is restricted to integer values, we perform a exhaustive search over its feasible domain $[0, s_{\max}]$，where $s_{\max}= \sum\limits_{i=1}^{K} N_i$. For each candidate $s'$ in the search, we compute the optimal $\underline{\alpha}'$ under the constraint $\mathcal{A}(s')$, which is a $K\times K$ quadratic programming problem with respect to $\underline{\alpha}$
>
> $$
> \underline{\alpha}'=\arg\min\limits_{\underline{\alpha}\in\mathcal{A}(s')}\underline{\alpha}^T\Theta^T{J}({{\underline{\theta}}_0})\Theta\underline{\alpha}.
> $$
>
>
> The quadratic coefficient matrix in this optimization problem is given by $\Theta^\top J(\underline{\theta}_0) \Theta$. Since the Fisher information matrix $J(\underline{\theta}_0)$ is positive semi-definite, the quadratic coefficient matrix is also positive semi-definite. This guarantees the existence of a global optimal solution. After getting $s'$ and $\underline{\alpha}'$, the function is then evaluated at $(s', \underline{\alpha}')$.
>
> In brief, for each $s'$, we solve for the corresponding optimal $\underline{\alpha}'$, yielding a finite collection of candidate solutions $(s', \underline{\alpha}')$ and their associated objective function values.
> After completing the search, the optimal solution $(s^\*, \underline{\alpha}^\*)$ is chosen as the pair that achieves the lowest objective function values among all candidate $(s', \underline{\alpha}')$. Since the feasible set of $s$ is finite and enumerable, and for each fixed $s'$ the optimization over $\underline{\alpha}'$ has a solution, the overall optimization problem is guaranteed to have at least one global solution. Hence, the optimal pair $(s^\*, \underline{\alpha}^\*)$ exists.
>
> It is worth noting that, for the sake of computational efficiency, we do not exhaustively enumerate all possible values of $s$ in our experiments. Instead, we perform a grid search using 1000 uniformly spaced steps over the feasible range of $s$, where the number of steps is denoted as $stepnumber = 1000$. Experimental results in Section 5 demonstrate that this strategy does not compromise the effectiveness or stability of the method.
>
> Thank you again for your valuable insights. We kindly request that the reviewer considers these points when assigning the final scores.

---

> ### Comment · Reviewer_rZbt · 2025-08-06
> **Response to Author**
>
> Thank you for the detailed responses. I find the answers to Q2–Q4 satisfactory.
>
> Regarding Q1, I appreciate the explanation of the dynamic update strategy. However, I still have some concerns about the stability of the Fisher information estimate, particularly in the initial epoch when only a small number of target samples are used. Clarifying its robustness would further strengthen the method.
>
> That said, given that the empirical results are promising and my concerns in Q2–Q4 have been adequately addressed, I am willing to raise my score.

---

> ### Author Response · Authors · 2025-08-08
>
> Thank you for your thoughtful response and your positive recognition of our work. We will provide a clarification regarding your remaining concern on Q1.
>
>
> >Question: The robustness of the algorithm and the initial Fisher information estimation.
>
> We understand the reviewer’s concern regarding the impact of the variance of the initial Fisher information matrix on the robustness of our algorithm OTQMS. We believe that Sections 5.3 and 5.4 help clarify this issue. As discussed in Section 5.4, we evaluate the performance of OTQMS under varying shot settings, which correspond to different quantities of target samples. A higher shot setting leads to better initial estimates of both the target model parameters and the Fisher information matrix. Nevertheless, OTQMS demonstrates stable performance and consistently outperforms the baselines across most shot settings, including those with very limited target samples, highlighting its robustness to the quality of the initial Fisher information. This robustness is largely attributable to the proposed dynamic strategy, which incorporates additional source samples to assist in training the Fisher information matrix in subsequent epochs, and whose effectiveness is further validated in Section 5.3.
>
>
> To further support the robustness of our algorithm, we supplement the main result table (Table 2 in the paper) by reporting the standard deviations of OTQMS and selected baselines, computed over 10 runs with different random seeds. Since each seed results in a different randomly sampled 10-shot target dataset, the relatively small standard deviations suggest that OTQMS remains robust despite variations arising from different target-data samplings and corresponding initial Fisher information estimates.
>
> **Performance on the Office-Home dataset under the same setting as Table 2 in the paper.** The horizontal axis lists the target domains, where the arrow notation (e.g., →Ar) indicates transferring **to** that domain from all others. The vertical axis lists the compared methods.
>
>
> |     **Method**      |    **→Ar**    |    **→Cl**    |    **→Pr**    |    **→Rw**    |  **Avg**  |
> |:-------------------:|:-------------:|:-------------:|:-------------:|:-------------:|:--------:|
> |   Target-Only       |  40.0 ± 0.8   |  33.3 ± 0.9   |  54.9 ± 0.7   |  52.6 ± 0.6   |  45.2    |
> | AllSources∪Target   |  77.0 ± 0.6   |  62.3 ± 0.7   |  84.9 ± 0.5   |  84.5 ± 0.4   |  77.2    |
> |   OTQMS (Ours)      |  78.1 ± 0.4   |  64.5 ± 0.5   |  85.2 ± 0.3   |  84.9 ± 0.3   |  78.2    |
>
>
> **Performance on the DomainNet dataset under the same setting of Table 2 in the paper**
>
>
> |     **Method**      |    **→C**    |    **→I**    |    **→P**    |    **→Q**    |    **→R**    |    **→S**    |  **Avg**  |
> |:-------------------:|:------------:|:------------:|:------------:|:------------:|:------------:|:------------:|:--------:|
> |   Target-Only       | 14.2 ± 1.1   |  3.3 ± 1.0   | 23.2 ± 1.3   |  7.2 ± 1.2   | 41.4 ± 1.0   | 10.6 ± 0.9   |  16.7    |
> | AllSources∪Target   | 71.7 ± 0.9   | 32.4 ± 1.2   | 60.0 ± 1.0   | 31.4 ± 1.1   | 71.7 ± 0.8   | 58.5 ± 0.8   |  54.3    |
> |   OTQMS (Ours)      | 72.8 ± 0.6   | 33.8 ± 0.8   | 61.2 ± 0.7   | 33.8 ± 0.8   | 73.2 ± 0.6   | 59.8 ± 0.6   |  55.8    |
>
>
> Thank you again for your thoughtful and constructive feedback. We kindly hope these clarifications demonstrate the robustness of our algorithm. We are truly grateful for your willingness to raise your score. Considering that the discussion phase is about to end, we would like to kindly remind you that—if you have no further concerns—you may click the “Edit” button at the top of your initial review to update your final score.

---

### Note · Authors · 2025-08-13

We sincerely thank the AC and reviewers for their time and constructive feedback throughout the review period, during which we carefully addressed all concerns point-by-point. We believe that the clarifications and additional analyses have further reinforced the soundness and validity of our main contributions.

In this Final Remark, we would like to restate our **motivation and novelty**. Our motivation arises from the observation—validated by Figure 1, Table 2, and further clarified in our response to Reviewer rZbt’s Q2—that using all available source samples in transfer learning may not yield optimal performance and can substantially increase training time. This motivates the question: *what is the optimal transfer quantity from each source task needed in training the target task?* As Reviewer 43HZ stated, *“The paper addresses a critical and often overlooked question in transfer learning: not just which sources to use, but how much data to use from each.”*

**Contributions – Our main contributions are as follows:**

1. **Unified theoretical framework** – We present a framework for determining the optimal transfer quantity of each source task. Specifically, we introduce a K-L divergence measure as the measure of generalization error. This measure is then minimized via asymptotic analysis to derive the optimal transfer quantity for each source task. As Reviewer ZUcy commented, *“The task that estimates the optimal quantity of source samples holds broad application scenarios. The derivation of the method is reasonable and the theoretical results are intuitive yet not trivial.”*

2. **Practical and efficient algorithm** – Based on the framework, we propose **OTQMS**, an architecture-agnostic and data-efficient algorithm for target model training in multi-source transfer learning. In particular, we design a *dynamic strategy* to enhance the robustness of the algorithm, as detailed in Section 4.3 and in our response to Reviewer rZbt’s Q1, which has been positively recognized by the reviewers. This algorithm works with both full model training and parameter-efficient methods like LoRA (Section 5.6).

3. **Extensive experimental studies** – We evaluate OTQMS on multi-source transfer learning tasks using the Office-Home and DomainNet datasets. The results demonstrate that OTQMS achieves significant improvements in both accuracy (Section 5.2) and data efficiency (Section 5.5), validating its effectiveness and robustness across diverse scenarios.

---

### Decision · Program_Chairs · 2025-09-17

**Decision:**

Accept (poster)

**Comment:**

This paper received generally positive ratings (5, 5, 4, 4). The reviewers appreciated the novel problem formulation of optimizing transfer quantities from each source task in multi-source transfer learning, with a theoretically-grounded KL-divergence framework. The main concerns centered on restrictive theoretical assumptions (small parameter distances between tasks, asymptotic analysis), misleading "high-dimensional" terminology, and the simplicity of the pooled MLE approach compared to modern transfer learning paradigms. The authors provided comprehensive rebuttals addressing most concerns, including computational complexity analysis, clarification of assumptions, and commitment to terminology corrections. All reviewers maintained or raised their scores post-rebuttal, acknowledging that while theoretical limitations exist, the practical contributions and consistent empirical improvements justify acceptance. Given the consensus from four reviewers supporting acceptance, the paper's novel contribution to an underexplored problem, and good empirical validation across multiple settings, the AC recommends acceptance. The authors should address remaining concerns in the final version, particularly clarifying the scope of theoretical assumptions and correcting the "high-dimensional" terminology.